# Efficient Patch Search in Whole Slide Images via Morphological Momentum Prototype Learning

## Abstract

Digital histopathology images play a crucial role in cancer diagnosis, therapeutic response prediction, and identification of clinically relevant morphological features. However, processing Whole Slide Images (WSI) with gigapixel resolution introduces significant challenges in computer vision, exceeding the memory capacity of standard vision encoders. To address this, recent methods employ a multi-stage pipeline: dissecting the image into small patches, extracting patch-level features, and aggregating these features using global pooling through Multi-Instance Learning (MIL) to form a final slide-level representation. Despite achieving clinical-grade performance, this approach becomes increasingly complex with higher magnification due to the quadratic increase in patch numbers and the generation of numerous irrelevant or redundant patches. This complexity burdens the global pooling network, resulting in long inference times and excessive computational resources, while redundant patches introduce noise during the MIL process, limiting the model's ability to utilize high-magnification features fully. To overcome these challenges, we propose **M**omentum **M**orphological **P**rototype **L**earning (MMPL), an efficient method that redefines WSI diagnosis as a searching process of relevant patch-level representations with a learned set of global prototypes. MMPL trains a fixed set of prototypes to retrieve the most informative patches, computing the diagnostic score using only the retrieved patches. Evaluated on WSI classification benchmarks, MMPL achieves state-of-the-art performance across various pathology tasks, including metastasis detection, tumor grading, and tumor subtyping.

## 1 Introduction

Digital histopathology images are widely used for cancer diagnosis Bejnordi et al. (2017); Bulten et al. (2022), therapeutic response prediction Bera et al. (2019); Niazi et al. (2019), and identifying clinically relevant morphological features Beck et al. (2011); Yamamoto et al. (2019). However, Whole Slide Images (WSIs), often examined at high magnification to capture subtle tissue symptoms, can reach gigapixel resolution (up to $150000 \times 150000$ pixels Zhang et al. (2022a)), exceeding the memory limits of standard vision encoders. Recent methods Lu et al. (2021) address this via a multi-stage pipeline: i) segmenting into patches, ii) extracting patch features, and iii) aggregating them with MIL for slide-level representation. Yet, as magnification increases, the number of patches grows quadratically, with many being irrelevant or redundant, leading to high computational cost and hindering advanced training schemes such as end-to-end learning. To address these limitations, prior works have proposed various sampling-based methods to focus on a more relevant subset of patches. However, these methods suffer from fundamental weaknesses. First, random or attention-based sampling fails to provide a reliable foundation for WSI learning. Random selection indiscriminately includes irrelevant background regions, while attention-based scores are often unstable in early training, leading to the inclusion of non-diagnostic patches. Since the final model updates are conducted through global pooling that operates independently of the sampling process, these strategies create a clear misalignment between the sampling criteria and the update objectives, which prevents the model from learning in a fully consistent manner. Second, prototype-based methods attempt to sample more informative patches via k-means clustering or GMM assignments,

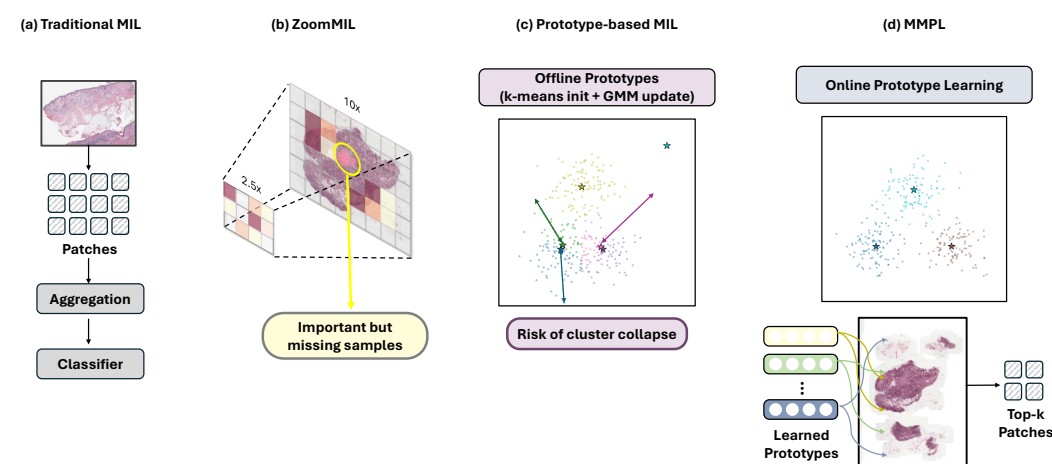

Figure 1: Comparison of existing methods for Whole Slide Image (WSI) analysis and our proposed MMPL. (a) Traditional MIL methods struggle with the large number of WSI patches. (b) Two-resolution method Thandiackal et al. (2022) can reduce memory usage, but it still suffers from the dilemma of low-resolution selection, which is often inaccurate if high-resolution features are not apparent at low magnification, such regions cannot be sampled. (c) Prototype-based methods Song et al. (2024) require fixed clustering and parametric refinement. (d) MMPL learns prototypes unsupervisedly and retrieves only a small, relevant subset.

but they also fall short. Clustering-based approaches select patches by nearest-prototype similarity, yet prototypes are updated by simple feature averaging—again, a disconnect between selection and learning. GMM-based alternatives provide soft assignments that are better aligned, but remain highly sensitive to the initial number of clusters and are prone to prototype collapse and redundancy, where several prototypes converge to the same mode while others remain unused. This not only destabilizes training but also leaves entire semantic concepts in WSIs underrepresented.

In this work, we propose MMPL (**M**omentum **M**orphological **P**rototype **L**earning), a framework that unifies patch selection and prototype learning through an optimal transport formulation with the Sinkhorn-Knopp algorithm. This enables non-collapsing prototypes that uniformly capture diverse morphological patterns, while ensuring that patch selection and prototype updates are coherently aligned to focus on the most informative regions. To further enhance generalization, a feature queue incorporates context from previous WSIs, yielding more robust prototype learning.

Our contributions are as follows:

1. A unified learning framework that directly aligns patch sampling and prototype learning objectives, overcoming the misalignment inherent in previous methods.

2. An optimal transport-based prototype learning method that effectively prevents prototype collapse and ensures a diverse, non-redundant representation of semantic patterns.

3. The efficient end-to-end pipeline is achieved by retrieving the most informative top-$k$ patches, reducing computational overhead while simultaneously improving performance.

On various WSI classification benchmarks for tumor subtyping, grading, and metastasis detection, MMPL consistently achieves state-of-the-art performance in both accuracy and efficiency, validating the effectiveness of our approach.

## 2 RELATED WORKS

### 2.1 MULTIPLE INSTANCE LEARNING

Multiple Instance Learning (MIL) is widely used for WSI analysis and can be approached explicitly—aggregating patch-level predictions via methods like Mean or Max-pooling (Campanella et al.,

2019; Zhang et al., 2022a)—or implicitly, using neural networks to combine patch embeddings. (Campanella et al., 2019; Zhang et al., 2022a) applied a Recurrent Neural Network (RNN) (Campanella et al., 2019) for slide-level aggregation. Attention-based MIL (AB-MIL) (Ilse et al., 2018) improves flexibility by weighting patch representations. CLAM (Lu et al., 2021) introduces an auxiliary task to evaluate instance relevance via attention size. Trans-MIL (Shao et al., 2021) models inter-patch relationships using Self-Attention, with efficiency improvements via linear variants. DSMIL (Li et al., 2021)incorporates multi-scale information, while HIPT (Chen et al., 2022) and HIGT (Guo et al., 2023) develop hierarchical representations. To mitigate overfitting, DTFD-MIL (Zhang et al., 2022a) generates sub-bags through instance resampling. Recent works enable fine-tuning via MC sampling from pre-trained encoders (Li et al., 2023), albeit with a two-stage pipeline. IBMIL (Lin et al., 2023) leverages K-means-based prototypes for retraining the aggregator.

## 2.2 Instance Selection Methods for WSIs

Traditional MIL models with global pooling are inefficient and sensitive to noise, as they process all patches equally. To improve this, methods such as random sampling (Lerousseau et al., 2021), reinforcement learning (Dong et al., 2018), and attention-based selection (BenTaieb & Hamarneh, 2018) have been proposed, though often requiring costly supervision. Others use iterative attention (Katharopoulos & Fleuret, 2019; Kong & Henao, 2022) or multi-scale zooming (Thandiackal et al., 2022). Prototype-based approaches (Yao et al., 2020; Lin et al., 2023; Song et al., 2024) group similar patches but rely on static clusters, limiting flexibility. In contrast, MMPL dynamically updates trainable prototypes using a momentum queue and retrieves informative patches via a shared similarity matrix, enabling efficient and adaptive selection.

## 2.3 Multi-vector Retrieval

Dense retrieval models often struggle with representing queries and documents as single vectors. To overcome this, methods like Polyencoder (Humeau et al., 2019), MEBERT (Luan et al., 2021), and MVR (Zhang et al., 2022b) use multiple embeddings, while token-level approaches such as Col-BERT (Khattab & Zaharia, 2020) store individual token embeddings for finer retrieval. Extensions like ALIGNER (Qian et al., 2022), COIL (Gao et al., 2021), and CITADEL (Li et al., 2022) improve alignment and efficiency. In this work, we reformulate WSI diagnosis as a multi-vector retrieval task, using global prototypes to retrieve informative patch-level features for slide-level prediction.

## 2.4 Self-Supervised Stabilization

Self-Supervised Learning techniques, such as MoCo (He et al., 2020), highlighted the necessity of maintaining a large feature queue of past features, while both MoCo and BYOL (Grill et al., 2020) emphasized using an Exponential Moving Average (EMA) mechanism to update the target encoder and ensure stable representations. In particular, MoCo demonstrated the importance of a memory queue for effective contrastive learning, while BYOL highlighted the role of an EMA-stabilized target network in preventing representation collapse. In this work, we adopt an EMA-based momentum encoder and feature queue to stabilize optimal-transport-based clustering of WSI patch features into global prototypes, yielding stable prototypes for top-$k$ patch retrieval under end-to-end training.

## 2.5 Optimal Transport

Clustering-based self-supervised learning methods generate supervision signals by iteratively grouping similar features, but they often suffer from prototype collapse, which degrades representation quality. To address this issue, prior work has introduced optimal-transport(OT)-based solutions: SeLa (Asano et al., 2019) uses the Sinkhorn–Knopp algorithm to enforce an equipartition constraint that ensures full prototype utilization, and SwAV (Caron et al., 2020) applies this algorithm in an online clustering setting to prevent collapse. In this work, we use OT-based clustering on WSI patch features to learn global morphological prototypes that directly guide dynamic top-$k$ patch retrieval for slide-level classification.

# 3 METHOD

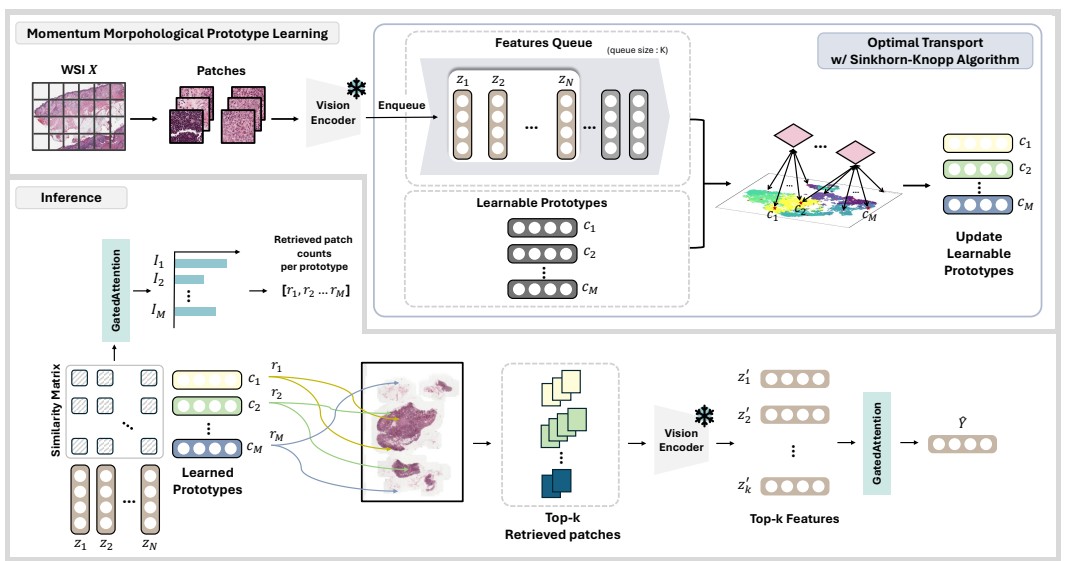

Figure 2: Overview of the MMPL framework. Prototypes are learned via optimal transport–based online clustering and used to dynamically select the top-$k$ most relevant patches. The vision encoder is jointly trained and recomputes features for these patches, while an EMA mechanism stabilizes feature extraction by smoothly aligning updates with the encoder.

We present **M**omentum **M**orphological **P**rototype **L**earning, a prototype-driven framework for WSI analysis that integrates optimal transport with dynamic patch retrieval (Figure 2). Within the MIL setting, only a subset of informative patches is sampled and aggregated through gated-attention pooling (Section 3.1). To incorporate broader context, patch features are stored in a momentum-based queue that stabilizes prototype learning (Section 3.2). Global prototypes are optimized via optimal transport to enforce balanced assignments and avoid collapse, and then used as queries to dynamically retrieve the most relevant top-$k$ patches for slide-level prediction (Section 3.3). The entire framework is trained end-to-end with a joint objective combining supervised classification and self-supervised prototype learning (Section 3.4).

## 3.1 SAMPLING WSI PATCH INSTANCES

Given a WSI $X$, the slide-level prediction $\hat{Y}$ is generated by training a classifier $f(X; \theta)$. Due to the high resolution of WSIs, $X$ is divided into $N$ non-overlapping smaller instances $X = \{x_1, \ldots, x_N\}$, where each patch $x_i \in \mathbb{R}^{W \times H \times 3}$. The slide-level prediction $\hat{Y}$ is then obtained through a global-pooling operation over instance-level latent labels $\hat{y}_i$, defined as:

$$\hat{Y} = \max\{\hat{y}_1, \ldots, \hat{y}_N\}, \quad \hat{y}_i \in \mathbb{R}. \tag{1}$$

Under WSI-level supervision, instance labels $\hat{y}_i$ are unavailable. Conventional approaches reformulate this problem as a multi-instance learning (MIL) task through the following steps: i) Images are encoded into feature representations $\mathcal{Z} = \{\mathbf{z}_1, \ldots, \mathbf{z}_N\}$ using a backbone $h$, where: $\mathbf{z}_i = h(x_i; \theta_1), \quad \mathbf{z}_i \in \mathbb{R}^d$. Here, $h$ can be any architecture, such as a CNN or Vision Transformer (ViT), parameterized by $\theta_1$. ii) Patch-level features $\mathcal{Z}$ are aggregated to compute the slide-level prediction:

$$\hat{Y} = g(\mathcal{Z}; \theta_2), \tag{2}$$

where $g(\cdot)$ is an attention-based pooling function followed by a linear classifier:

$$g(\mathcal{Z}; \theta_2) = \sigma\left(\sum_{i=1}^{N} a_i \mathbf{z}_i\right), \tag{3}$$

where $a_i$ represents the attention weight, computed as $a_i = \frac{\exp(\mathbf{w}^T \mathbf{z}_i)}{\sum_{j=1}^{N} \exp(\mathbf{w}^T \mathbf{z}_j)}$, $\sum_{i=1}^{N} a_i = 1$. Here, $\mathbf{w}$ is a learnable parameter that determines the importance of each patch feature, and $\sigma(\cdot)$ is a linear classifier. To reduce computational costs, traditional approaches train $\theta_1$ and $\theta_2$ separately: (1) initializing $\theta_1$ with a pretrained model and (2) freezing $\theta_1$ while learning $\theta_2$ under slide-level supervision.

Encoding all patches is computationally intensive, while random sampling reduces cost but risks missing essential information. Thus, Equation (3) becomes:

$$g(\mathcal{Z}; \theta_2) = \sigma\left(\sum_{i \in \mathcal{S}} \tilde{a}_i \mathbf{z}_i\right), \qquad (4)$$

where $\tilde{a}_i = \frac{\exp(\mathbf{w}^T \mathbf{z}_i)}{\sum_{j \in \mathcal{S}} \exp(\mathbf{w}^T \mathbf{z}_j)}$, $\sum_{i \in \mathcal{S}} \tilde{a}_i = 1$. and $\mathcal{S}$ denotes the index set of informative patch-level instances. This sampling process optimizes memory usage, facilitating end-to-end training with the visual encoders.

### 3.2 MOMENTUM MORPHOLOGICAL PROTOTYPE LEARNING

To learn a set of global morphological prototypes that represent shared patterns across WSIs, we propose a framework that combines a momentum queue with an optimal transport-based online clustering strategy. Our approach dynamically aligns patch features to learnable prototypes in a way that facilitates stable and consistent prototype learning throughout training.

**Feature Queue** To encourage global context sharing beyond the current WSI, we maintain a feature queue $\hat{\mathbf{Z}} = [\hat{\mathbf{z}}_1, \ldots, \hat{\mathbf{z}}_K]$ updated in a first-in-first-out (FIFO) manner, where $N \leq K$ that stores patch features extracted from previously processed slides. This design allows the prototypes to reflect global morphological patterns rather than being biased toward the localized distribution of the current slide. Consequently, it helps mitigate overfitting to slide-specific features and encourages the learning of more diverse and transferable prototype representations. In the end-to-end training setup, to stabilize the learning of prototypes, we populate the feature queue using outputs from a momentum encoder based on an exponential moving average (EMA) of the online encoder. This helps reduce cluster drift caused by rapidly updating encoder parameters.

**Prototype Learning via Optimal Transport** We adopt an optimal transport (OT) formulation with uniform marginal constraints to learn M global prototypes $\mathbf{C} = [\mathbf{c}_1, \ldots, \mathbf{c}_M] \in \mathbb{R}^{M \times d}$, which guarantees balanced representation of distinct morphological patterns while preventing prototype collapse and ensuring diverse information distribution.

These prototypes are trained dynamically using:

$$\mathcal{L}_{\text{prototype}} = -\sum_{i=1}^{N} \sum_{j=1}^{M} Q_{ij} \log P_{ij}, \qquad (5)$$

where $P_{ij} = \frac{\exp(\mathbf{z}_i^T \mathbf{c}_j / \tau)}{\sum_{k=1}^{M} \exp(\mathbf{z}_i^T \mathbf{c}_k / \tau)}$, and $Q$ represents pseudo-labels matrix for features $\mathbf{Z} = [\mathbf{z}_1, \ldots, \mathbf{z}_N]$. All vectors, including $\mathbf{z}_i$ and $\mathbf{c}_j$, are $\ell_2$-normalized ensuring computations are based on cosine similarity. We set the temperature $\tau = 0.1$ for all experiments.

To compute pseudo-labels for patch features online, we apply the Sinkhorn-Knopp algorithm Cuturi (2013) to minimize the transport distance between the entire feature queue $\hat{\mathbf{Z}}$ and the prototype set $\mathbf{C}$. This yields a transport matrix $\hat{Q} \in \mathbb{R}^{K \times M}$, computed by solving:

$$\max_{\hat{Q} \in \mathcal{Q}} \text{Tr}\left(\hat{Q}^T \hat{\mathbf{Z}} \mathbf{C}^T\right) + \epsilon H(\hat{Q}), \qquad (6)$$

where $H(\hat{Q}) = -\sum_{ij} \hat{Q}_{ij} \log \hat{Q}_{ij}$, and $\epsilon$ regularizes the entropy. The transport polytope $\mathcal{Q}$ is constrained as:

$$\mathcal{Q} = \left\{\hat{Q} \in \mathbb{R}_+^{K \times M} \;\middle|\; \hat{Q} \mathbf{1}_M = \frac{\mathbf{1}_K}{K}, \quad \hat{Q}^T \mathbf{1}_K = \frac{\mathbf{1}_M}{M}\right\}. \qquad (7)$$

Since the current batch features $\mathbf{Z} = [\mathbf{z}_1, \ldots, \mathbf{z}_N] \subset \hat{\mathbf{Z}}$, we extract the first $N$ rows from $\hat{Q}$, denoted as $Q_i = \hat{Q}_i^*$ for $\quad \forall\, 1 \leq i \leq N$. The resulting pseudo-labels matrix $Q$ is then used for prototype optimization through Equation (5).

Instead of computing OT within the current batch, we align features to prototypes using a entire feature queue, which stabilizes the assignment across iterations. While this alignment reflects the global morphological context, label assignments are limited to the current batch to ensure that prototype updates are guided by the local patterns of the current WSI.

### 3.3    PATCH SEARCH VIA MOMENTUM PROTOTYPES

Using the learned global prototype set $\mathbf{C}$, we dynamically retrieve $k$ relevant patch features $\mathbf{Z}'$ from the WSI feature set $\mathbf{Z}$. For a WSI $X$ divided into $N$ non-overlapping patches, we define the sampling index set $\mathcal{S}$ in Equation (4) as a query-based search, where prototypes from $\mathbf{C}$ act as queries to find the top-$k$ most relevant patch instances. Each prototype defines a corresponding sampling set of the most significant patches based on their relevance scores calculated using cosine similarity.

**Dynamic top-$k$ Search for MMPL.**    The top-$k$ set for each $\mathcal{S}_j$ is dynamically adjusted using attention weights $\mathbf{I} \in \mathbb{R}^M$, learned through downstream tasks. These weights, computed via a gated attention layer, determine the importance of each prototype and allocate candidates accordingly. This dynamic assignment allows the model to retrieve a varying number of samples for each prototype, enabling it to focus on collecting more task-relevant patches compared to uniform feature retrieval.

The refined sampling set is:

$$\mathcal{S}_j = \text{TopK}_{\lceil k \times \mathbf{I}_j \rceil} \left( \mathbf{Z}^T \mathbf{c}_j \right), \tag{8}$$

where $\mathbf{I}_j$ reflects the significance of prototype $\mathbf{c}_j$, guiding the distribution of candidates.

### 3.4    TRAINING MMPL

Training for MMPL incorporates both supervised and self-supervised objectives to effectively learn global prototypes and optimize classification performance. Given the slide-level label $Y$, the loss consists of the supervised log-loss for $\hat{Y}$, defined as $\mathcal{L}_{\text{CE}} = -\sum_i Y_i \log \hat{Y}_i$, and self-supervised prototype learning loss $\mathcal{L}_{\text{prototype}}$ from Equation (5). A weighting factor $\lambda$ balances the contribution of these two components, resulting in the overall loss:

$$\mathcal{L} = \mathcal{L}_{\text{CE}} + \lambda \mathcal{L}_{\text{prototype}}, \tag{9}$$

and we set $\lambda = 0.1$ for all experiments.

For comprehensive evaluation, MMPL is tested in two scenarios: (1) freezing the backbone parameters $\theta_1$ of the vision encoder $f_v$ while training only $\theta_2$, to ensure comparability with other WSI-MIL methods, and (2) enabling full end-to-end online training to jointly optimize all parameters.

**Online End-to-End Training.**    The combination of lightspeed transport and top-$k$ sampling reduces memory overhead, enabling efficient online end-to-end training. MMPL generates self-supervised labels by using the OT assignment distribution $Q_{i,:}$ as soft pseudo-labels for each patch, allowing gradients to flow directly through the vision encoder for joint optimization without pre-training. To prevent representation collapse, an Exponential Moving Average (EMA) stabilizes the retrieval feature extractor, ensuring robust and consistent representations.

### 3.5    INFERENCE MMPL

In the end-to-end setting, given a WSI $X$ with patch set $\{x_i\}_{i=1}^N$, these features are computed as $\mathbf{z}_i$ using the momentum (EMA) encoder, without any parameter updates. The resulting features and the learned prototypes $\mathbf{C} = [\mathbf{c}_1, \ldots, \mathbf{c}_M]$ are $\ell_2$-normalized, and we compute cosine similarities to obtain a similarity matrix between patches and prototypes. A gated-attention layer then produces prototype coefficients $\mathbf{I} \in \mathbb{R}^M$, which determine how the top-$k$ retrieved patches are distributed across prototypes and define variable-size index sets $\{S_j\}_{j=1}^M$.

| Model | CAMELYON16 | | | PANDA(K) | PANDA(R) | TCGA (standalone) | | |
|---|---|---|---|---|---|---|---|---|
| | Acc | WF1 | AUC | Kappa | | Acc | WF1 | AUC |
| ABMIL$_{(20\times)}$ | 0.847 ($\pm$0.008) | 0.844 ($\pm$0.008) | 0.887 ($\pm$0.013) | 0.835 ($\pm$0.005) | 0.820 ($\pm$0.015) | 0.821 ($\pm$0.006) | 0.821 ($\pm$0.006) | 0.894 ($\pm$0.002) |
| DSMIL$_{(10\times\rightarrow20\times)}$ | 0.817 ($\pm$0.015) | 0.812 ($\pm$0.015) | 0.848 ($\pm$0.016) | 0.754 ($\pm$0.023) | 0.791 ($\pm$0.026) | 0.837 ($\pm$0.008) | 0.837 ($\pm$0.008) | 0.925 ($\pm$0.003) |
| ZoomMIL$_{(10\times\rightarrow20\times)}$ | 0.860 ($\pm$0.008) | 0.858 ($\pm$0.008) | 0.869 ($\pm$0.009) | **0.860** ($\pm$0.008) | 0.811 ($\pm$0.018) | 0.835 ($\pm$0.010) | 0.834 ($\pm$0.010) | 0.912 ($\pm$0.007) |
| IBMIL$_{(20\times, \text{abmil})}$ | 0.865 ($\pm$0.008) | 0.863 ($\pm$0.009) | 0.898 ($\pm$0.006) | 0.414 ($\pm$0.204) | 0.456 ($\pm$0.214) | 0.851 ($\pm$0.009) | 0.851 ($\pm$0.010) | 0.932 ($\pm$0.002) |
| PANTHER$_{(20\times, \text{linear})}$ | 0.754 ($\pm$0.028) | 0.741 ($\pm$0.020) | 0.782 ($\pm$0.032) | 0.582 ($\pm$0.000) | 0.756 ($\pm$0.000) | 0.756 ($\pm$0.002) | 0.754 ($\pm$0.002) | 0.857 ($\pm$0.000) |
| PANTHER$_{(20\times, \text{mlp})}$ | 0.792 ($\pm$0.034) | 0.786 ($\pm$0.033) | 0.829 ($\pm$0.037) | 0.695 ($\pm$0.021) | **0.887** ($\pm$0.010) | 0.831 ($\pm$0.006) | 0.830 ($\pm$0.006) | 0.926 ($\pm$0.001) |
| MMPL$_{(20\times)}$ | **0.885** ($\pm$0.011) | **0.885** ($\pm$0.010) | **0.924** ($\pm$0.011) | 0.838 ($\pm$0.011) | 0.841 ($\pm$0.039) | **0.869** ($\pm$0.008) | **0.869** ($\pm$0.008) | **0.946** ($\pm$0.006) |

Table 1: Performance evaluation of the two-step approach using ResNet50 on the CAMELYON16, PANDA, and TCGA (standalone) datasets.

For each prototype $\mathbf{c}_j$, we allocate

$$r_j := |S_j| = \lceil k\, I_j \rceil$$

patches (as depicted in Figure 2), with $\sum_{j=1}^{M} r_j = k$, and keep only the top-$r_j$ patches with the highest similarity scores, optionally recomputing their representations with the encoder in the end-to-end setting. We then aggregate the retrieved features $\mathbf{Z}' = \{\mathbf{z}_s\}_{s \in S}$ (where $S = \bigcup_j S_j$ and $|S| = k$) via gated-attention pooling $g$ followed by the classifier $\sigma$ to obtain the slide-level prediction $\hat{Y}$.

For notational convenience, let $S = \{s_1, \ldots, s_k\}$ be an enumeration of the index set $S$, and write

$$\mathbf{Z}' = [\mathbf{z}'_1, \ldots, \mathbf{z}'_k],$$

where $\mathbf{z}'_\ell := \mathbf{z}_{s_\ell}$ for $\ell = 1, \ldots, k$.

## 4 EXPERIMENTS

### 4.1 DATASETS

Following previous works, we used three popular WSI datasets. (1) CAMELYON16 Bejnordi et al. (2017) is a widely used benchmark dataset from the Cancer Metastases in Lymph Nodes Challenge, aimed at detecting metastatic breast cancer. The dataset consists of 399 WSIs of lymph node biopsies (collected from two independent institutions), with annotated regions indicating cancer presence. (2) PANDA Bulten et al. (2022) is a large-scale benchmark dataset designed for the Prostate Cancer Grade Assessment Challenge. The dataset comprises an extensive collection of 9,555 Whole Slide Images (WSIs), which were collected from two distinct institutions: the Karolinska Institute (4,990 WSIs) and the Radboud University Medical Center (4,565 WSIs). This task, a 6-class classification based on the International Society of Urological Pathology (ISUP) grade, is evaluated primarily using Quadratic Weighted Kappa $\kappa$ to assess performance. (3) TCGA Tomczak et al. (2015), the subset of The Cancer Genome Atlas dedicated to lung cancer, includes data from two subtypes, namely Lung Adenocarcinoma and Lung Squamous Cell Carcinoma. The dataset consists of 1,042 WSIs containing 530 WSIs of LUAD and 512 WSIs of LUSC.

### 4.2 BASELINES

We compare MMPL with state-of-the-art MIL methods. Specifically, we consider ABMIL Ilse et al. (2018), which employs gated-attention pooling, and DSMIL Li et al. (2021), a multi-magnification approach encoding all patches across scales. Additionally, we evaluate ZoomMIL Thandiackal et al.

| | CAMELYON16 | | |
|---|---|---|---|
| **Model** | **ACC** | **WF1** | **AUC** |
| ABMIL$_{(20\times)}$ | $0.976_{\pm 0.006}$ | $0.976_{\pm 0.006}$ | $0.986_{\pm 0.006}$ |
| DSMIL$_{(10\times \to 20\times)}$ | $0.938_{\pm 0.053}$ | $0.938_{\pm 0.053}$ | $0.963_{\pm 0.045}$ |
| ZoomMIL$_{(10\times \to 20\times)}$ | $0.963_{\pm 0.008}$ | $0.963_{\pm 0.008}$ | $0.983_{\pm 0.008}$ |
| IBMIL$_{(20\times, \text{abmil})}$ | $0.977_{\pm 0.002}$ | $0.977_{\pm 0.002}$ | $0.992_{\pm 0.003}$ |
| PANTHER$_{(20\times, \text{linear})}$ | $0.773_{\pm 0.003}$ | $0.770_{\pm 0.003}$ | $0.797_{\pm 0.001}$ |
| PANTHER$_{(20\times, \text{mlp})}$ | $0.787_{\pm 0.006}$ | $0.786_{\pm 0.007}$ | $0.822_{\pm 0.008}$ |
| **MMPL** *(Ours)* $_{(20\times)}$ | $\mathbf{0.987}_{\pm \mathbf{0.005}}$ | $\mathbf{0.987}_{\pm \mathbf{0.005}}$ | $\mathbf{0.999}_{\pm \mathbf{0.001}}$ |

Table 2: Performance evaluation of the two-step approach using UNI on the CAMELYON16 dataset.

| | CAMELYON16 | | |
|---|---|---|---|
| **Model** | **ACC** | **WF1** | **AUC** |
| VIB$_{(20\times)}$ | $0.870_{\pm 0.009}$ | $0.869_{\pm 0.010}$ | $0.928_{\pm 0.013}$ |
| **MMPL** *(Ours)* $_{(20\times)}$ | $\mathbf{0.916}_{\pm \mathbf{0.014}}$ | $\mathbf{0.916}_{\pm \mathbf{0.014}}$ | $\mathbf{0.976}_{\pm \mathbf{0.009}}$ |

Table 3: A comparison of VIB and MMPL (ours) of the end-to-end approach using ResNet50 on the CAMELYON16 dataset.

(2022), IBMIL Lin et al. (2023), and PANTHER Song et al. (2024). ZoomMIL extends the multi-magnification paradigm by employing a hierarchical top-$k$ sampling strategy to select informative regions across scales. IBMIL uses K-means clustering to generate prototypes after training an aggregator, while PANTHER leverages a Gaussian Mixture Model (GMM) for unsupervised prototype-based learning. We also assess our sampling method in an end-to-end setting, where VIB Li et al. (2023) applies gated attention-based Markov sampling (see Table 3). For additional baselines not covered in the main text, please refer to the supplementary material.

### 4.3 RESULTS

We primarily use ResNet50 as the main encoder and additionally report UNI results for CAME-LYON16, as other encoders did not show clear gains. Models are evaluated using Accuracy (ACC), Weighted F1 (WF1), and ROC AUC (AUC). Table 1 presents results with ResNet50 on CAME-LYON16, PANDA, and TCGA, while Table 2 and Table 3 report UNI and end-to-end ResNet50 baselines, respectively. Additional comparisons, including VIB vs. MMPL with UNI and results with other encoders, are provided in the supplementary material. We also evaluate PANTHER Song et al. (2024) under default settings, training on 1M patches and 16 prototypes with K-means cluster-ing.

**CAMELYON16.** MMPL achieves state-of-the-art results across all metrics. With the ResNet50 encoder (see Table 1), MMPL outperforms ZoomMIL by 2.5% in accuracy and 5.5% in AUC, and exceeds IBMIL by 2.0% in accuracy and 2.6% in AUC. With the UNI encoder (see Table 2), MMPL surpasses ZoomMIL by 2.5% in weighted F1, 2.5% in accuracy, and 1.6% in AUC, using only a single resolution ($20\times$). It also outperforms IBMIL by 1.0% in both accuracy and weighted F1, and 0.7% in AUC. While UNI approaches ceiling-level performance, MMPL still provides meaningful improvements. Notably, MMPL yields larger gains with ResNet50, demonstrating robustness to weaker encoders and consistency across architectures.

**PANDA.** Results corresponding to the Karolinska Institute cohort are denoted as PANDA(K), and those from the Radboud University Medical Center cohort are denoted as PANDA(R). Our proposed MMPL achieves the highest Kappa score on PANDA(R), demonstrating its effectiveness in this evaluation setting (see Table 1). Specifically, MMPL outperforms ZoomMIL by 2.7% in Kappa(R), achieving the best performance among all baseline models. On PANDA(K), MMPL attains a Kappa score within 2.2% of the best-performing model, indicating comparable performance while main-

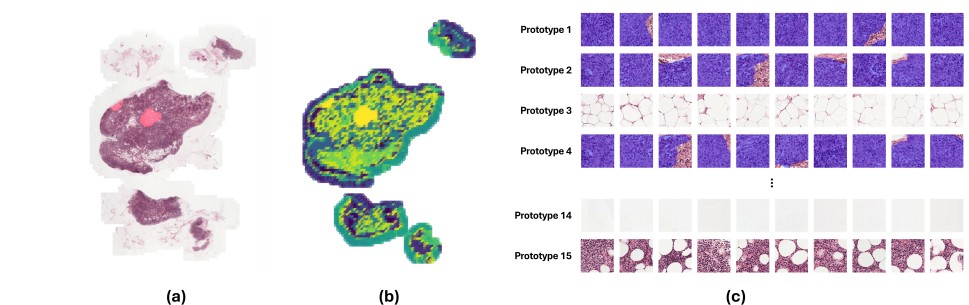

(a)           (b)           (c)

Figure 3: Visualization of a tumor slide image from the CAMELYON16 dataset, highlighting tumor regions and the patch visualizations for each prototype ($c_j, j = 1, 2, \cdots, M$). (a) The highlighted red regions indicate the human-annotated tumor segment. (b) The localization map highlights the significance of each prototype through intensity levels. (c) The blue area represents the annotated tumor mask, and prototypes are ranked in descending order of their importance scores, with higher-ranked prototypes capturing more critical tumor regions.

taining stability. These results indicate that MMPL provides robust and consistent predictions across different evaluation settings.

**TCGA.** We evaluated our method under two settings: standard 5-fold cross-validation (5-fold CV) and standalone test evaluation (5-fold CV-ST) using cross-validated models. The complete tables, including detailed 5-fold CV results, are provided in the supplementary material. In 5-fold CV, results were averaged over all five folds. In the standalone setting, we trained five models via 5-fold CV on the training set and aggregated their predictions on the test set via majority voting. Results from this setting are denoted as "standalone" in the corresponding table. As shown in Table 1, MMPL outperforms all baselines, achieving state-of-the-art performance across all metrics. Notably, it surpasses ZoomMIL by $3.4\%$ in accuracy and $3.4\%$ in AUC, despite using only a single resolution. Similarly, MMPL exceeds IBMIL by $1.8\%$ in accuracy and $1.4\%$ in AUC.

## 5 ANALYSIS

### 5.1 QUALITATIVE ANALYSIS

Figure 3 visualizes patches retrieved by each prototype ($c_j, j = 1, 2, ..., M$) for a CAMELYON16 tumor slide. In Figure 3(a), bright red marks human-annotated tumor regions, while Figure 3(b) shows the spatial distribution of patches colored by prototype importance (dark blue to yellow). Figure 3(c) ranks prototypes by normalized coefficients $\alpha_j$, which indicate each prototype's contribution to the slide-level prediction $\hat{Y}$. Higher-ranked prototypes capture distinctive cellular patterns and receive more patches, whereas less informative prototypes contribute less. This weighting ensures the aggregated representation emphasizes the most relevant features, improving both performance and interpretability.

### 5.2 ABLATION STUDY

**Feature Queue.** We evaluated the impact of the feature queue and found that its removal consistently degraded performance across all metrics (Table 4), confirming its effectiveness. By storing patch features from previous slides, the queue enables prototypes to capture diverse morphological patterns beyond the current slide, mitigating overfitting and improving generalization. A momentum encoder further stabilizes prototype learning by reducing fluctuations in the feature space, leading to more consistent and reliable cluster assignments.

**Momentum Morphological Prototype.** To assess the role of prototypes in our framework, we perform an ablation study comparing three configurations (see Table 4): a variant without prototypes, a uniform top-k assignment, and our dynamic top-k approach. The prototype-free model

| MMPL | CAMELYON16 | | |
|------|-----|-----|-----|
| | ACC | WF1 | AUC |
| **w/o queue** | $0.745_{\pm 0.064}$ | $0.743_{\pm 0.064}$ | $0.764_{\pm 0.078}$ |
| **w/o prototype** | $0.779_{\pm 0.036}$ | $0.775_{\pm 0.036}$ | $0.812_{\pm 0.030}$ |
| **uniform top-k** | $0.846_{\pm 0.057}$ | $0.845_{\pm 0.058}$ | $0.864_{\pm 0.056}$ |
| **K-means** | $0.812_{\pm 0.031}$ | $0.812_{\pm 0.032}$ | $0.865_{\pm 0.016}$ |
| *(Ours)* | $\mathbf{0.885}_{\pm 0.011}$ | $\mathbf{0.885}_{\pm 0.010}$ | $\mathbf{0.924}_{\pm 0.011}$ |

Table 4: Ablation study evaluating the effectiveness of queue, prototypes and clustering method using ResNet50 on the CAMELYON16. Full results, including additional encoders, are in the supplementary material.

yields the lowest performance, confirming the importance of prototype-based clustering in capturing informative patch representations via optimal transport. The uniform top-k strategy also underperforms compared to the dynamic variant, underscoring the benefit of selectively assigning patches to semantically relevant prototypes.

**Optimal Transport.** Building on the importance of prototypes, we compare our Sinkhorn-Knopp–based online clustering with a K-means baseline (see Table 4). The proposed method consistently outperforms the baseline, highlighting the limitations of K-means clustering, where prototype updates are constrained by fixed initial assignments and limited adaptability during training. In contrast, online clustering enables dynamic updates aligned with the evolving feature space, yielding more robust and discriminative representations.

**Comparison with previous end-to-end training.** Previous end-to-end methods include VIB Li et al. (2023), where the training procedure consist of two stages where the first stage learns the gated attention involving the entire patches and the second stage markov samples from the patches according to their relative attention score. Though being memory efficient, this procedure does not allow the sampling attention to be modified during the training of visual encoders, thus resulting in suboptimal performance. Table 3 shows the comparison of our MMPL with the current end-to-end training method in literature. It can be observed that in the scenario for end-to-end training including the parameters of the visual encoders, MMPL outperforms VIB in a significant magnitude.

## 6 CONCLUSION

In this work, we present **M**omentum **M**orphological **P**rototype **L**earning (MMPL), which reformulates WSI-MIL as multi-vector retrieval of patch representations via global prototypes. By selectively aggregating prototype-retrieved patches, MMPL streamlines pooling and achieves state-of-the-art performance on tasks such as metastasis detection, tumor grading, and subtyping. The architecture also supports end-to-end training, showing strong results in this fully trainable setting.

**Limitations and future work.** In this study, we showed that **M**omentum **M**orphological **P**rototype **L**earning supports both the traditional two-step pipeline and a fully end-to-end approach by selecting only the top-$k$ patches, substantially reducing the number of feature vectors processed compared to existing methods. Our experiments focus on slide-level classification tasks, as they provide explicit supervision on localized pathological signals in sparse tumor regions and constitute a stringent testbed for our prototype-based explicit patch selection mechanism. Building on this foundation, extending our framework beyond classification to more complex endpoints, such as prognosis and survival analysis that depend on patient-level and potentially multi-modal factors, is a natural direction for future work.

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
