# A APPENDIX

# B ALGORITHM AND IMPLEMENTATION

---

**Algorithm 1** Sinkhorn-Knopp Algorithm

---

1: **procedure** SINKHORN($\mathbf{L} \in \mathbb{R}^{m \times n}$, $\epsilon = 0.05$, $T = 3$)
2:                                                    ▷ Input: Logits matrix, regularization, iterations
3:      $\mathbf{Q} \leftarrow \exp(\mathbf{L}/\epsilon)^{\top}$
4:      $\mathbf{Q} \leftarrow \mathbf{Q}/\sum_{ij} \mathbf{Q}_{ij}$                                        ▷ Normalize all entries
5:      **for** $t = 1$ to $T$ **do**
6:          $\mathbf{Q} \leftarrow \mathbf{Q}/(\mathbf{Q}\mathbf{1}_n) \cdot m$                               ▷ Row normalization
7:          $\mathbf{Q} \leftarrow \mathbf{Q}/(\mathbf{1}_m\mathbf{Q}) \cdot n$                            ▷ Column normalization
8:      **end for**
9:      **return** $(\mathbf{Q} \cdot n)^{\top}$
10: **end procedure**

---

---

**Algorithm 2** MMPL Training Step

---

1: **procedure** MMPL
2: **Input** $\mathcal{D}$                                              ▷ set of WSI patches
3:      $\hat{Z}, C$                                           ▷ queue and prototypes
4:      $\theta_1, \theta_2, \theta_1'$                                      ▷ network parameters
5:      $h(\cdot; \theta_1), g(\cdot; \theta_2)$                             ▷ encoder and classifier
6:      $\lambda, \gamma$, optimizer                             ▷ update by gradients
7:      Sample $\{x_i\}, y \sim \mathcal{D}$
8:      **for each** $x_i$ **do**
9:          $z_i \leftarrow h(x_i; \theta_1')$                           ▷ use the momentum encoder
10:      **end for**
11:      $\hat{Z} \leftarrow \text{enqueue}(\hat{Z}, \{z_i\})$
12:      $\hat{Q}^* \leftarrow \text{SINKHORN}(\hat{Z}^{\top}C)$
13:      Compute $\mathcal{L}_{\text{prototype}}$ using Equation (5)
14:      Find $\{\mathcal{S}_j\}$ using Equation (8)
15:      **for** $i \in \cup_j \mathcal{S}_j$ **do**
16:          $z_i' \leftarrow h(x_i; \theta_1)$                            ▷ use the online encoder
17:      **end for**
18:      $\mathcal{L}_{\text{CE}} \leftarrow \text{cross\_entropy}(g(\{z_i'\}; \theta_2), y)$
19:      $\theta_1, \theta_2, C \leftarrow \text{optimizer}(\nabla(\mathcal{L}_{\text{CE}} + \lambda\mathcal{L}_{\text{prototype}}))$
20:      $\theta_1' \leftarrow \gamma\theta_1' + (1-\gamma)\theta_1$                     ▷ update by EMA
21:      **return** $\theta_1, \theta_2, \theta_1', C$
22: **end procedure**

---

# C TRAINING DETAILS

We train our model using the Adam optimizer with separate learning rates: $1 \times 10^{-4}$ for the multi-instance learning (MIL) component and $1 \times 10^{-5}$ for the encoder. Both learning rates are reduced by a factor of 0.5 following 10 epochs without improvement in validation loss. We trained the model for 100 epochs, employing validation set weighted F1 scores for model checkpoint selection, which were subsequently utilized for final evaluation on the test set. Our architecture comprises an input dimension of 1024, a hidden dimension of 512, and an attention dimension of 256, with a dropout rate of 0.25. For prototype learning, we employ 15 prototypes and retrieve the top 100 candidates during training. The queue size is set to 100,000 for CAMELYON16 and PANDA dataset, and 150,000 for TCGA dataset. We use a temperature of 0.1 to control output distribution sharpness, with a prototype learning loss weight of 0.1.

| | | CAMELYON16 | | |
|---|---|---|---|---|
| Encoder | Model | Acc | WF1 | AUC |
| ResNet18 | C2C | - | - | 0.911 |
| | ACMIL | - | $0.798\pm0.029$ | $0.841\pm0.081$ |
| ResNet50 | MAXMIL | $0.671\pm0.009$ | $0.640\pm0.030$ | $0.696\pm0.014$ |
| | MEANMIL | $0.659\pm0.016$ | $0.635\pm0.011$ | $0.664\pm0.017$ |
| | SparseConvMIL | $0.687\pm0.001$ | $0.677\pm0.006$ | $0.689\pm0.004$ |
| | CLAM-SB | $0.840\pm0.013$ | $0.832\pm0.015$ | $0.872\pm0.011$ |
| | TRANSMIL | $0.853\pm0.019$ | $0.836\pm0.026$ | $0.884\pm0.013$ |
| | MSMIL | $0.824\pm0.010$ | $0.814\pm0.011$ | $0.835\pm0.012$ |
| | ICMIL(maxpooling) | 0.843 | 0.796 | 0.874 |
| | ICMIL(abmil) | 0.882 | 0.819 | 0.924 |
| | ICMIL(DTFD-MIL) | **0.913** | 0.882 | 0.950 |
| | DGR-MIL | **0.917** | **0.913** | **0.957** |
| ResNet50 | **MMPL (Ours)** (20×) | $0.885\pm0.011$ | $\mathbf{0.885}\pm0.010$ | $\mathbf{0.924}\pm0.011$ |

Table 5: Performance evaluation of the two-step approach using ResNet18 and ResNet50 on the CAMELYON16 dataset.

# D ADDITIONAL EXPERIMENTS

## D.1 COMAPARATIVE EVALUATION OF MODEL VARIANTS

**Additional Baseline Comparisons.** In addition to the models introduced in the main text, we also provide a comparison with several additional models.

We compare with Cluster-to-Conquer Sharma et al. (2021) (referred to C2C), which is cluster-based retrieval method and CLAM Lu et al. (2021), which extends ABMIL Ilse et al. (2018) with an instance-level clustering loss. Additionally, we compare two spatially-aware methods, namely, TransMIL Shao et al. (2021) which models instance-level dependencies using transformer-based pooling, and SparseConvMIL Lerousseau et al. (2021) which selects random subsets of patches and employs sparse convolutions for pooling. We further compare with multi-scale methods MSMIL, which is multi-magnification approaches that encode all patches in a WSI across multiple magnifications. For completeness, we also include vanilla MIL methods based on max-pooling (MaxMIL) and mean-pooling (MeanMIL), following SparseConvMIL's strategy of random patch selection Lerousseau et al. (2021). Furthermore, we compare with ICMIL Wang et al. (2024), which uses max-pooling, abmil, and DTFD-MIL. We also compare with ACMIL Zhang et al. (2024) and DGR-MIL Zhu et al. (2024), which are multiple prototype approaches.

In Table 5, the performance scores for C2C, ACMIL, ICMIL, and DGR-MIL are directly taken from the corresponding published papers. Especially, MMPL outperforms Cluster-to-Conquer by $1.5\%$ in AUC. Also, MMPL achieves a $10.9\%$ higher weighted-F1 score compared to ACMIL and an $8.1\%$ higher weighted-F1 score compared to ICMIL (with ABMIL). MMPL surpasses ACMIL by $10.9\%$ in F1-score and $9.9\%$ in AUC. It also outperforms ICMIL (with MaxPooling) by $11.2\%\%$ in F1-score, $5.0\%$ in accuracy, and $5.7\%$ in AUC, while achieving $8.1\%$ higher F1-score and $0.3\%$ greater accuracy than ICMIL (with ABMIL) with the same AUC. MMPL achieves superior performance compared to most baselines, with the exception of ICMIL(with DTFD-MIL) and DGR-MIL. While these models report slightly higher scores, they also utilize significantly more patches per WSI. This suggests that MMPL provides a more computationally efficient alternative while maintaining competitive performance.

**Encoder Architecture Variants.** Table 6 presents the performance of our model with various encoders, including ViT-L/16 DINOv2 Chen et al. (2024) (referred to as UNI), which was pre-trained on a large histopathology dataset (100M images, 100K WSIs), demonstrating that utilizing advanced encoders consistently enhances performance across all datasets. In addition to UNI, other advanced encoders such as CONCH (CONtrastive Learning from Captions for Histopathology) Lu et al. (2024), a vision-language foundation model for histopathology, pretrained on 1.17M

| | CAMELYON16 | | |
|---|---|---|---|
| **Encoder** | **Acc** | **WF1** | **AUC** |
| ResNet50 | 0.885±0.011 | 0.885±0.010 | 0.924±0.011 |
| UNI | 0.987±0.005 | 0.987±0.005 | 0.999±0.001 |
| CONCH | 0.988±0.004 | 0.988±0.004 | 0.996±0.002 |
| GigaPath | 0.986±0.004 | 0.986±0.004 | 0.995±0.002 |

Table 6: Performance evaluation of the Two-Step approach with Different Encoders on the CAME-LYON16 dataset.

| **Top-K** | **WF1** | **Acc** |
|---|---|---|
| 80 | 0.868±0.014 | 0.868 ±0.014 |
| **100** | **0.885**±0.010 | **0.885** ±0.011 |
| 120 | 0.860±0.040 | 0.861 ±0.040 |
| 150 | 0.850±0.044 | 0.851 ±0.044 |

Table 7: Ablation study on the effect of top-$k$ values in patch retrieval. Performance peaks at $k = 100$, with diminishing returns at higher values and lower accuracy at smaller $k$ due to insufficient patch selection.

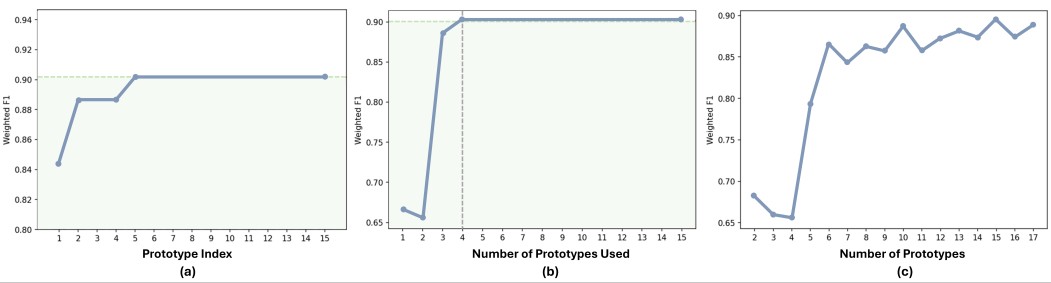

Figure 4: (a) Evaluated the model performance by progressively removing prototypes one by one. (b) Evaluated the model performance by progressively accumulating prototypes one by one. (c) The results of the experiments, where the total number of prototypes was varied as a hyperparameter.

hispathology-specific image-caption pairs, and Prov-GigaPath Xu et al. (2024) (referred to as Giga-Path), a whole-slide pathology foundation model pretrained on 1.3 billion image tiles from 171,189 slides, achieve similarly high performance. However, as their results are closely aligned with UNI, we report only UNI as the representative among the advanced encoders in our main analysis. While ResNet50 He et al. (2016), which was pre-trained on ImageNet but not specifically on WSIs, is a relatively lightweight model compared to advanced encoders such as UNI, CONCH, and GigaPath, our model still outperforms other baselines when using this encoder, demonstrating its robustness across different backbone architectures. (See Table 1.)

## D.2 PARAMETER SENSITIVITY ANALYSIS

**Number of Prototypes.** We computed importance scores for 15 prototypes using gated attention on the CAMELYON-16 dataset. Prototypes were ranked by their importance, reflecting their retrieval frequency. As shown in Figure 4(a), the top 4 prototypes contribute most to classification. It is evident that, beyond the 4th prototype, removing further prototypes has little impact on performance. Notably, eliminating the highest-ranked prototype significantly reduces performance, underscoring its importance. Figure 4(2) shows the results of sorting the learned prototypes and then accumulating them to investigate how many prototypes should be used. Again, we observe that using the top 4 prototypes restores the performance, confirming that these prototypes are essential for optimal classification. Figure 4(c) examines the effect of varying the total number of prototypes. Due to our dynamic top-$k$ assembly, highly relevant prototypes heavily influence MMPL's perfor-

| Encoder | MMPL | CAMELYON16 | | |
|---|---|---|---|---|
| | | ACC | WF1 | AUC |
| ResNet50 | w/o queue | $0.745\pm0.064$ | $0.743\pm0.064$ | $0.764\pm0.078$ |
| | w/o prototype | $0.779\pm0.036$ | $0.775\pm0.036$ | $0.812\pm0.030$ |
| | uniform top-k | $0.846\pm0.057$ | $0.845\pm0.058$ | $0.864\pm0.056$ |
| | K-means | $0.812\pm0.031$ | $0.812\pm0.032$ | $0.865\pm0.016$ |
| | *(Ours)* | $\mathbf{0.885\pm0.011}$ | $\mathbf{0.885\pm0.010}$ | $\mathbf{0.924\pm0.011}$ |
| UNI | w/o queue | $0.736\pm0.013$ | $0.729\pm0.012$ | $0.704\pm0.028$ |
| | w/o prototype | $0.846\pm0.018$ | $0.844\pm0.019$ | $0.874\pm0.018$ |
| | uniform top-k | $0.859\pm0.050$ | $0.859\pm0.050$ | $0.909\pm0.047$ |
| | K-means | $0.836\pm0.033$ | $0.836\pm0.034$ | $0.914\pm0.018$ |
| | *(Ours)* | $\mathbf{0.987\pm0.005}$ | $\mathbf{0.987\pm0.005}$ | $\mathbf{0.999\pm0.001}$ |

Table 8: Ablation study evaluating the effectiveness of queue, prototypes, and clustering method using ResNet50 and UNI on CAMELYON16.

mance. However, beyond a certain threshold, additional prototypes do NOT proportionally lead to improved performance.

**Number of Top-K.** In this ablation study, we investigate the impact of different values for the top-$k$ parameter in the patch retrieval process. As shown in Table 7, we evaluate four different values for $k$: 80, 100, 120, and 150, and report the corresponding weighted F1 scores (WF1) and accuracy (Acc). As observed, performance improves as we increase the $k$ from 80 to 100, with WF1 and accuracy reaching their peak at $k = 100$ ($88.5 \pm 1.0$ and $88.5 \pm 1.1$, respectively). However, as $k$ increases further to 120 and 150, performance declines, suggesting that retrieving too many patches leads to diminishing returns in both weighted F1 score and accuracy. This decline can be attributed to the fact that when $k$ is set too high, redundant or irrelevant patches may be retrieved, diluting the contribution of the most informative patches to the final prediction. A higher number of patches may introduce noise rather than enhancing the feature diversity captured by the model, as evidenced by the drop in performance when $k$ exceeds 100. Conversely, if top-$k$ is set too low, too few patches are selected, which may negatively affect the accuracy of the final prediction, as important features may be underrepresented. The optimal top-$k$ value appears to be 100, where the trade-off between retrieving enough patches to cover diverse features and avoiding redundancy is well balanced.

### D.3 ADDITIONAL ABLATION EXPERIMENTS

To evaluate the contribution of each component in our framework, we conduct comprehensive ablation studies using both ResNet50 and UNI encoders (See Table 8). The results consistently demonstrate that our method outperforms all ablated variants, validating the effectiveness of each proposed component—namely, the queue mechanism, prototype learning, and clustering strategy.

**Generalization Across Encoder Architectures.** Notably, our approach exhibits strong generalizability across encoder architectures. Despite the inherent performance advantage of UNI due to its superior representational capacity, the ablation results show that it still underperforms our full method in all cases. This highlights that even high-capacity encoders benefit substantially from the complete training framework, reinforcing the effectiveness of our proposed components.

**Analysis of Key Components and Design Choices.** Among the ablated variants, the models without the queue or prototype components consistently yield the lowest performance across metrics (See Table 8 first and second row of each encoders). This underscores the critical importance of these two components in our method, suggesting that both the memory-based representation and prototype learning are essential to achieving strong performance.

Furthermore, the variant employing a uniform allocation of top-$k$ patches per prototype—while still using both queue and prototype—performs worse than our dynamic allocation approach. This

| Model | CAMELYON16 | | | PANDA(K) | PANDA(R) | TCGA | | | TCGA (standalone) | | |
|---|---|---|---|---|---|---|---|---|---|---|---|
| | Acc | WF1 | AUC | Kappa | | Acc | WF1 | AUC | Acc | WF1 | AUC |
| ABMIL(20×) | 0.847 | 0.844 | 0.887 | 0.835 | 0.820 | 0.848 | 0.847 | 0.915 | 0.821 | 0.821 | 0.894 |
| | (±0.008) | (±0.008) | (±0.013) | (±0.005) | (±0.015) | (±0.004) | (±0.004) | (±0.002) | (±0.006) | (±0.006) | (±0.002) |
| DSMIL(10×→20×) | 0.817 | 0.812 | 0.848 | 0.754 | 0.791 | 0.864 | 0.864 | **0.937** | 0.837 | 0.837 | 0.925 |
| | (±0.015) | (±0.015) | (±0.016) | (±0.023) | (±0.026) | (±0.009) | (±0.009) | (±0.004) | (±0.008) | (±0.008) | (±0.003) |
| ZoomMIL(10×→20×) | 0.860 | 0.858 | 0.869 | **0.860** | 0.811 | 0.847 | 0.847 | 0.846 | 0.835 | 0.834 | 0.912 |
| | (±0.008) | (±0.008) | (±0.009) | (±0.008) | (±0.018) | (±0.007) | (±0.008) | (±0.010) | (±0.010) | (±0.010) | (±0.007) |
| IBMIL(20×, abmil) | 0.865 | 0.863 | 0.898 | 0.414 | 0.456 | 0.862 | 0.862 | 0.909 | 0.851 | 0.851 | 0.932 |
| | (±0.008) | (±0.009) | (±0.006) | (±0.204) | (±0.214) | (±0.001) | (±0.001) | (±0.003) | (±0.009) | (±0.010) | (±0.002) |
| PANTHER(20×, linear) | 0.754 | 0.741 | 0.782 | 0.582 | 0.756 | 0.788 | 0.788 | 0.873 | 0.756 | 0.754 | 0.857 |
| | (±0.028) | (±0.020) | (±0.032) | (±0.000) | (±0.000) | (±0.001) | (±0.001) | (±0.000) | (±0.002) | (±0.002) | (±0.000) |
| PANTHER(20×, mlp) | 0.792 | 0.786 | 0.829 | 0.695 | **0.887** | 0.826 | 0.826 | 0.916 | 0.831 | 0.830 | 0.926 |
| | (±0.034) | (±0.033) | (±0.037) | (±0.021) | (±0.010) | (±0.006) | (±0.006) | (±0.002) | (±0.006) | (±0.006) | (±0.001) |
| MMPL(20×) | **0.885** | **0.885** | **0.924** | 0.838 | 0.841 | **0.882** | **0.882** | 0.930 | **0.869** | **0.869** | **0.946** |
| | (±0.011) | (±0.010) | (±0.011) | (±0.011) | (±0.039) | (±0.008) | (±0.008) | (±0.005) | (±0.008) | (±0.008) | (±0.006) |

Table 9: Performance evaluation of the two-step approach using ResNet50 on the CAMELYON16, PANDA, and TCGA datasets.

| Model | CAMELYON16 | | | PANDA(K) | PANDA(R) | TCGA | | | TCGA (standalone) | | |
|---|---|---|---|---|---|---|---|---|---|---|---|
| | Acc | WF1 | AUC | Kappa | | Acc | WF1 | AUC | Acc | WF1 | AUC |
| ABMIL(20×) | 0.976 | 0.976 | 0.986 | 0.921 | 0.900 | 0.938 | 0.938 | 0.981 | 0.934 | 0.934 | 0.987 |
| | (±0.006) | (±0.006) | (±0.006) | (±0.011) | (±0.014) | (±0.005) | (±0.005) | (±0.003) | (±0.011) | (±0.011) | (±0.003) |
| DSMIL(10×→20×) | 0.938 | 0.938 | 0.963 | 0.889 | 0.871 | 0.926 | 0.926 | 0.978 | 0.922 | 0.922 | 0.922 |
| | (±0.053) | (±0.053) | (±0.045) | (±0.014) | (±0.014) | (±0.005) | (±0.005) | (±0.001) | (±0.004) | (±0.004) | (±0.004) |
| ZoomMIL(10×→20×) | 0.978 | 0.978 | 0.993 | 0.931 | 0.938 | 0.947 | 0.947 | 0.981 | 0.924 | 0.924 | 0.983 |
| | (±0.009) | (±0.009) | (±0.005) | (±0.008) | (±0.013) | (±0.002) | (±0.002) | (±0.001) | (±0.004) | (±0.004) | (±0.002) |
| IBMIL(20×, abmil) | 0.977 | 0.977 | 0.992 | 0.845 | 0.913 | 0.930 | 0.930 | 0.964 | 0.939 | 0.939 | 0.988 |
| | (±0.002) | (±0.002) | (±0.003) | (±0.036) | (±0.008) | (±0.007) | (±0.007) | (±0.008) | (±0.003) | (±0.003) | (±0.002) |
| PANTHER(20×, linear) | 0.773 | 0.770 | 0.797 | 0.866 | 0.909 | 0.933 | 0.933 | 0.978 | 0.942 | 0.942 | 0.988 |
| | (±0.003) | (±0.003) | (±0.001) | | | (±0.001) | (±0.001) | (±0.000) | (±0.006) | (±0.006) | (±0.001) |
| PANTHER(20×, mlp) | 0.787 | 0.786 | 0.822 | 0.923 | 0.931 | 0.934 | 0.934 | 0.982 | **0.949** | **0.949** | **0.992** |
| | (±0.006) | (±0.007) | (±0.008) | | | (±0.001) | (±0.001) | (±0.000) | (±0.003) | (±0.003) | (±0.000) |
| **MMPL (Ours)** (20×) | **0.987** | **0.987** | **0.999** | **0.934** | **0.948** | **0.948** | **0.984** | 0.907 | 0.935 | 0.935 | 0.981 |
| | (±0.005) | (±0.005) | (±0.001) | (±0.006) | (±0.004) | (±0.004) | (±0.003) | (±0.010) | | | |

Table 10: Performance evaluation of the two-step approach using UNI on the CAMELYON16, PANDA, and TCGA datasets.

supports the hypothesis that allowing the number of selected patches per prototype to adapt based on their relevance leads to more effective learning. In other words, the model benefits from assigning more patches to more important prototypes rather than enforcing uniformity, emphasizing the value of dynamic prototype-wise instance selection.

In the comparison of prototype assignment strategies, our method using Sinkhorn-Knopp optimal transport consistently outperforms the ablation with standard K-means assignment. These results demonstrate the effectiveness of balanced assignment constraints, where the uniform marginal constraints in optimal transport ensure that all prototypes are utilized equally, preventing prototype collapse that commonly occur in unconstrained clustering methods.

## D.4 FULL RESULTS OF MAIN EXPERIMENTS

The full results of the main experiments are provided in Tables 9–12.

| Model | CAMELYON16 | | | PANDA(K) | PANDA(R) | TCGA | | | TCGA (standalone) | | |
|---|---|---|---|---|---|---|---|---|---|---|---|
| | Acc | WF1 | AUC | Kappa | | Acc | WF1 | AUC | Acc | WF1 | AUC |
| VIB(20×) Li et al. (2023) | 0.870 (±0.009) | 0.869 (±0.010) | 0.928 (±0.013) | **0.921** (±0.007) | **0.894** (±0.004) | **0.909** (±0.003) | **0.909** (±0.003) | **0.967** (±0.000) | 0.866 (±0.008) | 0.866 (±0.008) | **0.947** (±0.002) |
| **MMPL *(Ours)*** (20×) | **0.916** (±0.014) | **0.916** (±0.014) | **0.976** (±0.009) | 0.861 | 0.843 | 0.898 (±0.020) | 0.898 (±0.020) | 0.954 (±0.007) | **0.869** (±0.008) | **0.869** (±0.008) | 0.939 |

Table 11: Performance evaluation of the end-to-end approach using ResNet50 on the CAME-LYON16, PANDA, and TCGA datasets.

| Model | CAMELYON16 | | | PANDA(K) | PANDA(R) | TCGA | | | TCGA (standalone) | | |
|---|---|---|---|---|---|---|---|---|---|---|---|
| | Acc | WF1 | AUC | Kappa | | Acc | WF1 | AUC | Acc | WF1 | AUC |
| VIB(20×) Li et al. (2023) | **0.990** (±0.006) | **0.990** (±0.006) | **0.999** (±0.000) | **0.943** (±0.006) | **0.949** (±0.008) | **0.955** (±0.003) | **0.955** (±0.003) | **0.988** (±0.001) | 0.936 (±0.008) | 0.935 (±0.008) | **0.986** (±0.002) |
| **MMPL *(Ours)*** (20×) | 0.985 | 0.985 | **0.999** | 0.918 | 0.894 | 0.940 (±0.010) | 0.941 (±0.010) | 0.981 (±0.005) | **0.939** | **0.939** | 0.939 |

Table 12: Performance evaluation of the end-to-end approach using UNI on the CAMELYON16, PANDA, and TCGA datasets.

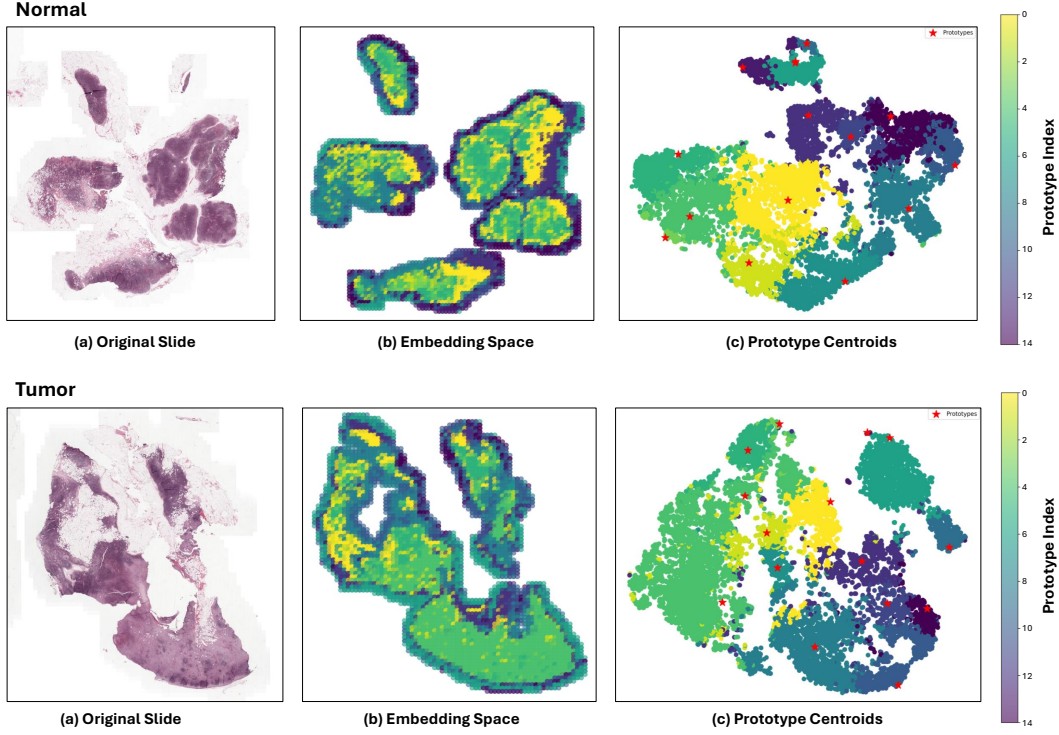

Figure 5: The top and bottom rows, respectively, represent images extracted from a normal slide and a tumor slide. In both rows, (a) shows the original slide, (b) depicts the embedding space of features, and (c) illustrates the patch distribution according to prototype centroids. The prototype indices are organized sequentially from 0 to 14 and are represented on a color gradient that progresses from yellow to purple. It's important to note that a prototype with an index of 0 indicates the highest importance score, suggesting it plays a critical role in distinguishing features within the dataset.

**A localization map visualization**

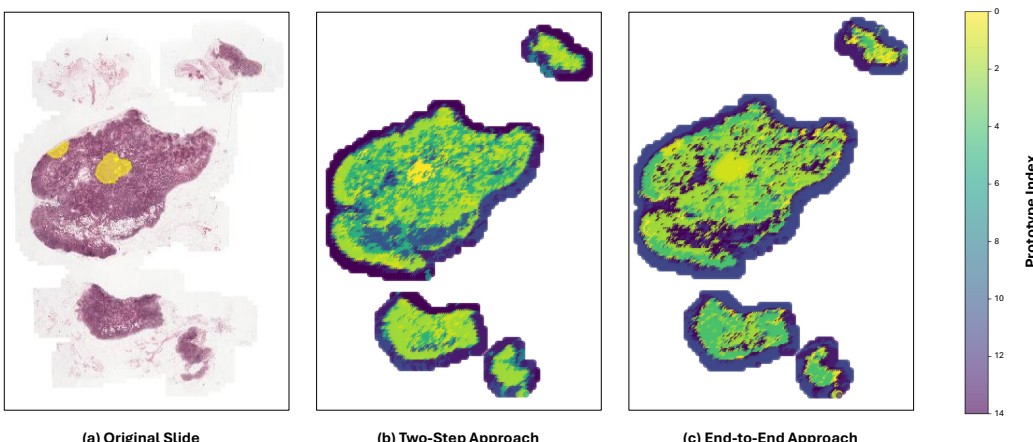

(a) Original Slide        (b) Two-Step Approach        (c) End-to-End Approach

Figure 6: A localization map visualization for a ResNet50-based MMPL on the Camelyon16 dataset, showing which prototype corresponds closely to each patch. (a) Original slide with the annotated tumor region highlighted in yellow. (b) Two-Step approach and (c) End-to-End approach, where colors from purple to yellow indicate increasing prototype importance. The End-to-End method exhibits better alignment with the annotated regions, including the smaller lesion on the left.