# OpenReview forum: "Efficient Patch Search in Whole Slide Images via Morphological Momentum Prototype Learning"
_ICLR.cc/2026/Conference — Submitted to ICLR 2026_

### Official Review · Reviewer_tnwR · 2025-10-22

**Soundness:** 3
**Presentation:** 3
**Contribution:** 3
**Rating:** 4
**Confidence:** 4

**Summary:**

This work presents an efficient method for WSI analysis, called Momentum Morphological Prototype Learning (MMPL). MMPL trains a fixed set of prototypes to retrieve the most informative patches, and then computes the diagnostic score using only the retrieved patches, which can largely reduce the computational cost. The authors conducted experiments on three public datasets to evaluate the proposed method, along with a comparison with existing methods. Ablation study is also carried out to verify the contribution of the key components of the proposed method. Experimental results show that MMPL achieves the best performance on the CAMELYON16 and TCGA datasets for metastasis detection and tumor subtyping, but has inferior performance on the PANDA dataset for tumor grading.

**Strengths:**

The manuscript presents a new paradigm for WSI analysis. Different from previous works that utilized all patches of WSI for analysis, this work proposed MMPL that trains several prototypes to retrieve the most informative patches for WSI representation. In this manner, the computational cost is significantly reduced. For prototyping learning, the authors adopted an optimal transport (OT) formulation with uniform marginal constraints to achieve balanced representation of distinct morphological patterns while preventing prototype collapse and ensuring diverse information distribution. In summary, the proposed method is of some novelty.

**Weaknesses:**

1)	The proposed method involves multiple hyperparameters, such as the number of prototypes, k value for top-k sampling, and K value for the feature queue. Although the authors conducted experiments to investigate the effects of prototype number and k value on the classification performance on CAMELYON16, it is still unclear whether the proposed method is sensitive to those hyperparameters across different datasets.
2)	The computational cost of retrieval should be considered during training. Given the queue size set to 100,000, there is nontrivial computation per training iteration. Please elaborate on it.
3)	The proposed method did not achieve state-of-the-art performance on the PANDA dataset.

**Questions:**

Besides the weakness mentioned above, there are some concerns as follows:
1)	Please specify the number of WSIs of the PANDA and TCGA datasets in Section 4.1.
2)	Please briefly introduce the workflow of inference after the introduction of Section 3.4, which could give readers a better understanding of the test stage.
3)	The authors mentioned that their method MMPL used only 1.18% of the patch embeddings. It is better to provide the computation process in the supplementary file.
4)	For the visualization of the prototypes in Figure 3, I am surprised to see that the white background (non-tissue regions, i.e., prototype 14) is accounted for in this work. Normally, we exclude the white background in the data pre-processing stage. Please elaborate on it.
5)	The proposed method did not achieve state-of-the-art performance on the PANDA dataset, and there is no analysis of the reason. Besides, I am not clear about PANDA (R) and PANDA (K). For the prostate cancer grading tasks, did the authors compute the kappa score between the model predictions and the ground truth labels? Accuracy is also a commonly used evaluation metric.

---

> ### Author Response · Authors · 2025-11-27
>
> We sincerely appreciate the reviewer’s positive evaluation of our work, and for the time you devoted to its evaluation. Below, we respond to each of the concerns and questions you have outlined.
>
> ---
>
> > The proposed method involves multiple hyperparameters, such as the number of prototypes, k value for top-k sampling, and K value for the feature queue. Although the authors conducted experiments to investigate the effects of prototype number and k value on the classification performance on CAMELYON16, it is still unclear whether the proposed method is sensitive to those hyperparameters across different datasets.
> >
>
> We agree that examining hyperparameter sensitivity beyond CAMELYON16 is important for assessing the robustness of the proposed method. To address this, we conducted additional ablation studies on the effects of queue size, the number of top-k patches, and the number of prototypes using ResNet50 features across PANDA and TCGA datasets.
>
> ### Feature Queue Size
>
> | Dataset | Camelyon16 |  |  |  |  | PANDA (K) | PANDA (R) |  |  | TCGA (5-fold-cv) |  |  |
> | --- | --- | --- | --- | --- | --- | --- | --- | --- | --- | --- | --- | --- |
> | queue_size | Acc | WF1 | AUC |  | queue_size | Kappa | Kappa |  | queue_size | Acc | WF1 | AUC |
> | 85000 | 0.854 | 0.853 | 0.880 |  | 25000 | 0.831 | 0.781 |  | **150000** | **0.882** | **0.885** | 0.924 |
> | 90000 | 0.853 | 0.851 | 0.876 |  | 50000 | 0.805 | 0.785 |  | 200000 | 0.875 | 0.875 | **0.925** |
> | **100000** | **0.885** | **0.885** | **0.924** |  | 70000 | 0.800 | 0.799 |  | 250000 | 0.879 | 0.878 | **0.925** |
> | 150000 | 0.795 | 0.793 | 0.798 |  | 90000 | 0.790 | 0.789 |  |  |  |  |  |
> | 200000 | 0.785 | 0.781 | 0.768 |  | **100000** | **0.838** | **0.841** |  |  |  |  |  |
> |  |  |  |  |  | 150000 | 0.826 | 0.802 |  |  |  |  |  |
>
> ### Top-k
>
> | Dataset | PANDA (K) | PANDA (R) |  | TCGA |  |  |
> | --- | --- | --- | --- | --- | --- | --- |
> | top_k | Kappa | Kappa | top_k | Acc | WF1 | AUC |
> | 80 | 0.817 | 0.759 | 80 | 0.871 | 0.870 | 0.929 |
> | **100** | **0.836** | **0.820** | **100** | **0.876** | **0.875** | **0.930** |
> | 120 | 0.820 | 0.812 | 120 | 0.856 | 0.857 | 0.923 |
> | 150 | 0.830 | 0.768 | 150 | 0.871 | 0.871 | 0.908 |
>
> ### Number of Prototypes
>
> | Dataset | PANDA (K) | PANDA (R) |  | TCGA |  |  |
> | --- | --- | --- | --- | --- | --- | --- |
> | num_proto | Kappa | Kappa | top_k | Acc | WF1 | AUC |
> | 3 | 0.814 | 0.775 | 3 | 0.780 | 0.780 | 0.890 |
> | 4 | 0.847 | 0.796 | 4 | 0.837 | 0.837 | 0.898 |
> | **5** | **0.875** | **0.836** | 5 | 0.847 | 0.847 | 0.914 |
> | 10 | 0.802 | 0.740 | 10 | 0.828 | 0.828 | 0.889 |
> | 15 | 0.836 | 0.820 | **15** | **0.876** | **0.875** | **0.930** |
> | 17 | 0.818 | 0.802 | 17 | 0.866 | 0.866 | 0.923 |
>
> Overall, the results indicate that the appropriate choice of the number of prototypes $M$ varies by dataset and task. On PANDA, for example, values of $M$ other than 15 yield superior performance, suggesting that further performance gains are possible with more precise tuning on this dataset.
>
> > The computational cost of retrieval should be considered during training. Given the queue size set to 100,000, there is nontrivial computation per training iteration. Please elaborate on it.
> >
>
> We understand the concern regarding retrieval overhead during training. To quantify its impact, we measured the proportion of time spent on prototype retrieval and top-k aggregation within each training iteration. Using controlled experiments on the CAMELYON16 dataset with a ResNet50 backbone and the same GPU (NVIDIA GeForce RTX 3090), we found that the retrieval and top-k selection stage accounts for 8.89% of the total training time. This indicates that, although the queue size is large, the associated computational cost represents a relatively small portion of the overall training budget.

---

> ### Author Response · Authors · 2025-11-27
>
> > The proposed method did not achieve state-of-the-art performance on the PANDA dataset.
> >
>
> We have empirically observed that achieving high performance on both the K subset and the R subset of the PANDA dataset is challenging. This pattern is consistent with the behavior of other baseline models, which also show a noticeable performance gap between K and R. We assume that this is because the R and K subsets follow different distributions due to differences in scanners and staining protocols, which increases the difficulty of achieving uniformly strong performance across both domains.
>
> In addition, through the extended hyperparameter search presented earlier, we identified a configuration under which our method performs competitively. Specifically, when the number of prototypes is set to 5 for the K subset, our model achieves a kappa score of 0.875 on the K subset, which is comparable to previously reported state-of-the-art results. Although the values initially reported in the manuscript were obtained under limited tuning, the existence of such a configuration demonstrates that the proposed method is capable of attaining state-of-the-art performance on PANDA when appropriate hyperparameter choices are made.
>
> > - Please specify the number of WSIs of the PANDA and TCGA datasets in Section 4.1.
> > - Please briefly introduce the workflow of inference after the introduction of Section 3.4, which could give readers a better understanding of the test stage.
> >
>
> We appreciate these helpful suggestions and have incorporated them into the revised manuscript. Specifically, we now explicitly state the number of WSIs used for the PANDA and TCGA datasets in Section 4.1, and we have added a brief description of the inference workflow immediately after Section 3.4. These revisions have improved the clarity of the manuscript.
>
> > The authors mentioned that their method MMPL used only 1.18% of the patch embeddings. It is better to provide the computation process in the supplementary file.
> >
>
> Thank you for raising this point and for suggesting to include the computation process. We acknowledge and have corrected a related error in Section 4.3 (Results for Camelyon16): the phrase "Using only 1.18% of patch embeddings," was meant to describe the patch usage for the E2E vision encoder training but was mistakenly inserted into the discussion of the two-step results. This has also been corrected.
>
> > For the visualization of the prototypes in Figure 3, I am surprised to see that the white background (non-tissue regions, i.e., prototype 14) is accounted for in this work. Normally, we exclude the white background in the data pre-processing stage. Please elaborate on it.
> >
>
> Regarding the observation about the white background prototype, our data pre-processing followed the same pipeline as ZoomMIL, which applies background filtering during patch extraction. In practice some patches from white regions within the WSI, such as small non-tissue areas, were still extracted because the filtering was not fully exhaustive. As a result, a prototype representing this white background (prototype 14) appears in Figure 3 even though background patches were filtered during pre-processing.
>
> > I am not clear about PANDA (R) and PANDA (K). For the prostate cancer grading tasks, did the authors compute the kappa score between the model predictions and the ground truth labels? Accuracy is also a commonly used evaluation metric.
> >
>
> We apologize for the confusion caused by this point. We inadvertently omitted the meanings of PANDA-K and PANDA-R in the manuscript. PANDA is the dataset released as part of the Prostate Cancer Grade Assessment Challenge and includes two cohorts collected at the Karolinska Institute (PANDA-K) and the Radboud University Medical Center (PANDA-R). In our manuscript, “PANDA (K)” and “PANDA (R)” refer to the evaluation results for these two subsets. We have added a brief description of both cohorts in the revised version.
>
> For the prostate cancer grading task, we follow the Challenge protocol and compute the **Quadratic Weighted Kappa** between the model predictions and the ground truth grade labels. This metric is standard for the PANDA benchmark because it reflects the ordinal nature of the grading system. Previous studies using the PANDA dataset [1, 2] also adopt Quadratic Weighted Kappa as the primary evaluation metric. In line with these practices, we report kappa as the main metric for our PANDA experiments.
>
> ---
>
> Thank you for the valuable comments that guided this revision. We believe the changes and additional analyses respond to your concerns, and we welcome any further questions or suggestions.
>
> [1] Andrew H Song et al., Morphological prototyping for unsupervised slide representation learning in computational pathology.
>
> [2] Riddhasree Bhattacharyya et al., Efficient grading of prostate cancer WSI with deep learning.

---

### Official Review · Reviewer_KdfC · 2025-11-04

**Soundness:** 2
**Presentation:** 2
**Contribution:** 2
**Rating:** 4
**Confidence:** 3

**Summary:**

This paper proposes a novel framework named Momentum Morphological Prototype Learning (MMPL) for weakly supervised classification of Whole Slide Images (WSIs). The method addresses the computational inefficiency in multiple instance learning caused by an excessive number of instances by reformulating the problem as an efficient patch search and retrieval task. MMPL employs an optimal transport (OT) formulation to learn a set of prototypes, a design intended to prevent prototype collapse and ensure diversity in pattern representation. Additionally, the authors adopt a feature queue combined with exponential moving average (EMA) updates to mitigate cluster drift induced by rapid encoder parameter changes. Based on the similarity between prototypes and image patches, the method selects a subset from the WSI's patch collection and performs classification using only this subset. The authors demonstrate the effectiveness of their approach on multiple public histopathology datasets, and ablation studies confirm the contribution of each component within the proposed framework.

**Strengths:**

1. It raises a valuable question: how can we address the explosive growth in the number of patches during WSI tiling due to increased resolution, along with the consequent issues of computational efficiency and noise?
2. The proposed framework introduces an optimal transport (OT) formulation to learn prototypes in a self-supervised manner, and incorporates a retrieval mechanism to reduce computational cost during classification; furthermore, the entire framework is end-to-end trainable, demonstrating a certain degree of novelty.
3. The effectiveness of MMPL is validated on multiple datasets, with comparisons to various state-of-the-art methods, and ablation studies are conducted to verify the contribution of each component.

**Weaknesses:**

1. The authors' central claim—that their framework achieves high efficiency—has not been substantiated by sufficient experimental evidence or rigorous analysis.
2. The experimental comparisons exhibit anomalous results, which the authors fail to explain; these discrepancies may indicate incorrect implementation of the method or an unfair experimental setup.
3. Key techniques such as optimal transport for preventing prototype collapse, and the use of a feature queue combined with exponential moving average (EMA) to mitigate feature drift, have been previously proposed in influential works, yet the authors do not acknowledge or discuss these prior contributions.

**Questions:**

1. Could the authors provide a computational complexity analysis and performance metrics (e.g., inference latency, throughput) for the proposed framework when processing a single whole slide image (WSI) to completion of classification, and compare these metrics with those of other existing methods?
2. In Table 1, the Kappa score of IBMIL appears unusually low. Is this due to an implementation error? If not, could the authors provide an analysis explaining the underlying reasons?
3. Is it possible that end-to-end training, while effectively fitting the training task, might lead to overfitting and thus perform poorly on out-of-domain test sets or in real-world scenarios?

---

> ### Author Response · Authors · 2025-11-27
>
> We are grateful for the reviewer’s positive remarks, especially for highlighting our approach to handling patch explosion, the novelty of the OT-based prototype retrieval framework, and the effectiveness of MMPL shown in our experiments.
>
> ---
>
> > The authors' central claim—that their framework achieves high efficiency—has not been substantiated by sufficient experimental evidence or rigorous analysis.
> Could the authors provide a computational complexity analysis and performance metrics (e.g., inference latency, throughput) for the proposed framework when processing a single whole slide image (WSI) to completion of classification, and compare these metrics with those of other existing methods?
> >
>
> ### Computational Costs
>
> |  | MMPL(ours) | ABMIL | ZoomMIL |
> | --- | --- | --- | --- |
> | Throughput (slides/sec) | **6.03** | 5.1 | 1.28 |
> | Mem. Usage (Avg) | **90.96 MB** | 122.53 MB | 1007.79 MB |
> | Mem. Usage (Max) | **664.30 MB** | 691.41 MB | 7694.00 MB |
> | # Params | 796,419 | 788,739 | 1,839,365 |
> | FLOPs (N=10000) | **10.71G** | 15.75G | 31.73G |
> | Theoretical FLOPs | $Nd^2 + \dots$ | $\frac{3}{2}Nd^2 + \dots$ | $\frac{5}{2} Nd^2 + \dots$ |
>
> Efficiency regarding our framework is an important point, and we appreciate the opportunity to clarify it. We conducted controlled experiments to compare inference cost on the CAMELYON16 dataset. All models (MMPL, ABMIL, and ZoomMIL) were evaluated under identical conditions using a single CPU thread. The theoretical complexity analysis employs standard notation, with $N$ denoting the number of patches and $d$ the feature dimension.
>
> Regarding throughput, MMPL achieves the highest performance because classification is applied only to the top-k retrieved patches. For FLOPs measured at $N=10,000$ patches, MMPL is the most efficient at 10.71 GFLOPs, compared with 15.75 GFLOPs for ABMIL and 31.73 GFLOPs for ZoomMIL. The observed FLOPs ratio (MMPL:ABMIL:ZoomMIL ≈ 1:1.5:3) is consistent with the theoretical complexity ratio of 1:1.5:2.5, which supports the architectural efficiency of the proposed method.
>
> > The experimental comparisons exhibit anomalous results, which the authors fail to explain; these discrepancies may indicate incorrect implementation of the method or an unfair experimental setup.
> In Table 1, the Kappa score of IBMIL appears unusually low. Is this due to an implementation error? If not, could the authors provide an analysis explaining the underlying reasons?
> >
>
> We conducted additional experiments using the publicly available GitHub repository. Specifically, for the IBMIL model, we observed that the original implementation did not natively support multi-class classification for the PANDA dataset; therefore, we simply extended it to enable multi-class training.
> Furthermore, we adjusted the optimization strategy to better suit this multi-class setting: we replaced the original CosineAnnealingLR scheduler with ReduceLROnPlateau and performed slight adjustments to the learning rate.
>
> The results obtained from these experiments on the PANDA dataset with ResNet50 are as follows:
>
> - PANDA (K): mean kappa score 0.414 with a standard deviation of 0.204.
> - PANDA (R): mean kappa score 0.456 with a standard deviation of 0.214.
>
> These new results demonstrate the performance of IBMIL under our adjusted, multi-class appropriate settings. We have included the updated scores achieved through these slight adjustments in the revision.

---

> ### Author Response · Authors · 2025-11-27
>
> > Key techniques such as optimal transport for preventing prototype collapse, and the use of a feature queue combined with exponential moving average (EMA) to mitigate feature drift, have been previously proposed in influential works, yet the authors do not acknowledge or discuss these prior contributions.
> >
>
> We sincerely thank the reviewer for kindly pointing out the related work that we had overlooked. In the revised manuscript, we have addressed this in the Related Work section, explicitly acknowledging the prior contributions of Optimal Transport and the combined use of the Feature Queue with EMA for mitigating feature drift.
>
> > Is it possible that end-to-end training, while effectively fitting the training task, might lead to overfitting and thus perform poorly on out-of-domain test sets or in real-world scenarios?
> >
>
> We note that this concern is understandable. Indeed, the inherent sparsity and large scale of whole-slide image (WSI) data make cross-domain generalization a particularly challenging problem. Given that datasets such as CAMELYON16 (breast cancer metastasis detection) and TCGA-LUAD / TCGA-LUSC (lung cancer–related cohorts from The Cancer Genome Atlas) involve different tissue types and are acquired under varying staining characteristics and scanner settings, it is not necessarily expected that a vision backbone trained on one dataset would generalize well to another.
>
> Our method, therefore, does not aim to train a universal vision encoder; rather, it is designed to jointly adapt the vision encoder within a domain-specific MIL framework. In other words, the proposed end-to-end approach enables effective adaptation of the encoder to the specific histopathological characteristics of each dataset, rather than assuming strong cross-domain generalization.
>
> ---
>
> We are grateful for your careful evaluation. The revisions aim to respond directly to your observations, and we look forward to hearing whether they meet your expectations.

---

> > ### Comment · Reviewer_KdfC · 2025-11-28
> >
> > Thank you for the detailed response. While the clarifications are helpful, several key concerns remain.
> > 1. Regarding the computational complexity analysis, it seems that the authors only analyzed the aggregation network (the parameter count in the table is only ~800k). As far as I know, in WSI classification the main computational bottleneck is the feature extraction by the encoder. While MMPL does reduce the number of patches fed into the aggregation network via retrieval, it does not appear to reduce the time spent on feature extraction. This suggests that the efficiency gains of MMPL may have only a limited impact on the end-to-end inference time.
> > 2. For the updated IBMIL results on the PANDA dataset, the standard deviation remains unusually large (around 0.2, compared to roughly 0.02 for other methods). This still indicates a potentially unfair or unstable comparison.
> > 3. The authors describe MMPL as a domain-specific MIL framework, which is reasonable, but some level of generalization is still necessary for practical applicability. For example, it would strengthen the paper if the authors could report results where the model is trained on dataset A and tested on a different but related dataset B (for the same task but from a different source), in order to demonstrate cross-dataset generalization.

---

> > > ### Author Response · Authors · 2025-12-03
> > >
> > > > Regarding the computational complexity analysis, it seems that the authors only analyzed the aggregation network (the parameter count in the table is only ~800k). As far as I know, in WSI classification the main computational bottleneck is the feature extraction by the encoder. While MMPL does reduce the number of patches fed into the aggregation network via retrieval, it does not appear to reduce the time spent on feature extraction. This suggests that the efficiency gains of MMPL may have only a limited impact on the end-to-end inference time.
> > > >
> > >
> > > We concur with the reviewer's precise observation regarding the aggregation stage. Our MMPL utilizes the Top-K retrieval mechanism to reduce the number of patches input to the Aggregation Network, thereby effectively mitigating the computational load during the aggregation phase.
> > >
> > > However, we must clarify that the core goal of our End-to-End (E2E) contribution was not to fundamentally improve the computational speed of the feature extraction process itself. Instead, the primary focus was on leveraging this Top-K selection mechanism to construct a memory-efficient framework for the training process of the entire Vision Encoder, thereby enabling stable E2E learning. Our key contribution, therefore, lies in establishing an efficient, Top-K based E2E learning framework, rather than achieving a substantial reduction in the inference time of the feature extraction step.
> > >
> > > > For the updated IBMIL results on the PANDA dataset, the standard deviation remains unusually large (around 0.2, compared to roughly 0.02 for other methods). This still indicates a potentially unfair or unstable comparison.
> > > >
> > >
> > > We analyzed the results obtained by training the IBMIL model using the ResNet50 encoder, which revealed the following unstable characteristics. Crucially, we must explicitly state that this phenomenon of severe instability was not observed when training with the UNI encoder under the identical experimental setup.
> > >
> > > 1. Analysis of Instability with ResNet50
> > > Our experiments conducted with the ResNet50 backbone, utilizing 5 seeds, revealed the following behavior:
> > >
> > > - 4 out of 5 seeds: The model exhibited catastrophic learning failure, resulting in the complete inability to predict 3 out of the 6 total classes (Failure to Predict).
> > > - 1 seed: While yielding a relatively high score compared to the other seeds, this single run still displayed instability in the prediction of 1 class.
> > >
> > > This polarized outcome, in which 4 runs failed severely while 1 run succeeded moderately, artificially inflated the standard deviation of the average score. We conclude that this high std is the consequence of this specific learning failure tied to the ResNet50 backbone.
> > >
> > > 2. Validation with the UNI Encoder
> > > To confirm this diagnosis, we conducted experiments using the UNI encoder under the identical experimental setup.
> > > The results, which have been incorporated into Table 6 of the Supplementary Material, show significantly reduced variance:
> > >
> > > - PANDA (K) (kappa): $0.845 (\pm 0.036)$
> > > - PANDA (R) (kappa): $0.913 (\pm 0.008)$
> > >
> > > Crucially, with the UNI encoder, there was no instance of failure to predict any of the 6 classes across all 5 seeds. The observed std is now well within the stable range similar to that reported by other comparison methods. This evidence suggests that the high std was an artifact of the ResNet50's specific interaction with the IBMIL framework.

---

> ### Author Response · Authors · 2025-12-03
>
> > The authors describe MMPL as a domain-specific MIL framework, which is reasonable, but some level of generalization is still necessary for practical applicability. For example, it would strengthen the paper if the authors could report results where the model is trained on dataset A and tested on a different but related dataset B (for the same task but from a different source), in order to demonstrate cross-dataset generalization.
> >
>
> We performed a cross-dataset experiment by applying an MMPL model trained on the PANDA dataset (with a ResNet-50 backbone) to the CAMELYON16 dataset.
>
> | Method | Train dataset (source) | Test dataset (target) | Source score (in PANDA-K, Kappa) | Target score (WF1) |
> | --- | --- | --- | --- | --- |
> | E2E (optimized to source dataset) | PANDA | CAMELYON16 | 0.884 | 0.227 |
> | E2E (less-tuned, similar hyperparameters between source and target) | PANDA | CAMELYON16 | 0.817 | 0.529 |
> | Two-step (similar hyperparameters between source and target) | PANDA | CAMELYON16 | 0.836 | 0.423 |
>
> These results indicate that the configuration highly optimized for PANDA does not transfer well to CAMELYON16, as the hyperparameters that are optimal for PANDA differ from those appropriate for CAMELYON16. A less-tuned configuration with hyperparameters closer to those used on CAMELYON16 achieves better cross-dataset performance, though at the cost of reduced source performance. This behavior is consistent with the substantial domain shift between PANDA and CAMELYON16 (different tissue types, staining characteristics, scanners, and even clinical targets), and reflects a broader challenge of cross-dataset generalization in WSI rather than a limitation unique to MMPL.
>
> We believe this limitation reflects a trade-off inherent to achieving state-of-the-art performance in a specific, specialized domain: aggressive, dataset-specific tuning improves in-domain accuracy but typically reduces robustness under large domain shifts. We fully agree that stronger cross-dataset generalization is highly desirable, and we now explicitly acknowledge this as an important open direction. MMPL is designed as a domain-specific MIL framework, and we expect it to be complementary to future advances in domain-generalization and stain/center normalization for WSI.

---

### Official Review · Reviewer_6sEU · 2025-11-04

**Soundness:** 3
**Presentation:** 2
**Contribution:** 2
**Rating:** 4
**Confidence:** 4

**Summary:**

The paper proposes Momentum Morphological Prototype Learning (MMPL), a prototype-driven framework for efficient Whole Slide Image (WSI) classification. MMPL learns a fixed set of global morphological prototypes and uses an optimal transport assignment with the Sinkhorn-Knopp algorithm over a momentum feature queue to avoid prototype collapse and keep prototypes diverse, and retrieve only the most informative top-k patch features per slide. This retrieval reduces the number of patch vectors the aggregator must process and enables end-to-end training with lower memory use. The authors evaluate MMPL on standard WSI benchmarks (CAMELYON16, PANDA, TCGA), report state-of-the-art accuracy/AUC/Kappa in several settings, and present ablations showing benefits of the feature queue, OT-based prototypes, and dynamic top-k retrieval.

**Strengths:**

1 Reformulating WSI diagnosis as a multi-vector retrieval problem driven by learned prototypes and solving prototype collapse with an OT constraint is a fresh combination of ideas that hasn’t been widely applied in this context. Combining a momentum queue with Exponential Moving Average (EMA) for stable prototype learning is a reasonable adaptation and integration of ideas derived from self-supervised learning.

2 Experiments across three widely-used WSI datasets (CAMELYON16, PANDA, TCGA) with multiple baselines (ABMIL, DSMIL, ZoomMIL, IBMIL, PANTHER, VIB) show consistent improvements. The ablation (queue/prototypes/OT vs K-means/uniform top-k) supports the mechanistic claims.

3 The method is explained step-by-step with helpful figures showing architecture and a visualization of prototype patch retrieval. Loss terms and the joint training objective are clearly defined.

4 WSI analysis is a high-impact application area; achieving similar or better performance while dramatically reducing processed patches addresses a real practical constraint (compute/memory) and can enable wider deployment and more frequent end-to-end training.

**Weaknesses:**

1 Efficiency claims need quantitative runtime/memory ablation. The paper states that only 1.18% of patches are used and claims reduced inference time, but provides no systematic runtime / GPU memory / FLOPs measurements vs baselines.

2 Sensitivity to key hyperparameters (M, queue size, top-k policy, τ, λ). The method depends on prototype count M, the queue size K, and the top-k allocation vector Ij. Only limited ablation is shown.

3 Theoretical justification or failure modes. While OT with uniform marginals mitigates collapse, the paper lacks a short formal discussion of when prototype assignments might still be suboptimal (e.g., extremely class-imbalanced slides) and how the method handles rare morphologies.

4 The MMPL architecture diagram is not aesthetically pleasing, except for the defect of misaligned box lines at first glance. Some symbols do not appear in the main text and are not explained in the captions

**Questions:**

1 Runtime & memory: Please report GPU memory usage and per-slide inference latency for MMPL and for at least tow strong baseline on the same hardware.


2 Prototypes and M selection: How sensitive is performance to M? Despite OT uniform marginals, is there a risk that some prototypes remain unused? Provide the distribution of prototype assignments (counts per prototype) and show whether OT enforcement truly yields balanced semantic coverage.

3 Provide clearer details and results comparing the two-step frozen-backbone training to full end-to-end training. Because the paper claims end-to-end is possible, thus conducting more ablation studies would make the argument more convincing. If possible, Could you provide a short expert evaluation or annotation to show prototypes align with meaningful histology concepts?

---

> ### Author Response · Authors · 2025-11-27
>
> We greatly appreciate the reviewer's recognition that reformulating WSI diagnosis as a prototype-driven multi-vector retrieval problem with an OT constraint offers a fresh combination of ideas in this context. In what follows, we address the weaknesses and questions you have raised in detail.
>
> ---
>
> > Efficiency claims need quantitative runtime/memory ablation. The paper states that only 1.18% of patches are used and claims reduced inference time, but provides no systematic runtime / GPU memory / FLOPs measurements vs baselines.
> >
>
> > Please report GPU memory usage and per-slide inference latency for MMPL and for at least tow strong baseline on the same hardware.
> >
>
> ### Computational Costs
>
> |  | MMPL(ours) | ABMIL | ZoomMIL |
> | --- | --- | --- | --- |
> | Throughput (slides/sec) | **6.03** | 5.1 | 1.28 |
> | Mem. Usage (Avg) | **90.96 MB** | 122.53 MB | 1007.79 MB |
> | Mem. Usage (Max) | **664.30 MB** | 691.41 MB | 7694.00 MB |
> | # Params | 796,419 | 788,739 | 1,839,365 |
> | FLOPs (N=10000) | **10.71G** | 15.75G | 31.73G |
> | Theoretical FLOPs | $Nd^2 + \dots$ | $\frac{3}{2}Nd^2 + \dots$ | $\frac{5}{2} Nd^2 + \dots$ |
>
> We appreciate the reviewer's careful reading regarding the initial phrasing in our Introduction. We acknowledge the potential for misinterpretation regarding the initial statement on reduced inference time in the Introduction. We recognize that the original manuscript lacked sufficient empirical support for this specific claim, and for the sake of clarity and precision, the ambiguous sentence has been removed from the revised manuscript.
>
> We confirm that we do not explicitly claim reduced inference time in the main body of the paper. However, we emphasize that the removal of this statement does not imply reduced efficiency. On the contrary, **MMPL's inference time is demonstrably faster than baselines, as evidenced by the results of our experiments.** We apologize for the initial confusion.
>
> We conducted controlled experiments to compare inference cost on the CAMELYON16 dataset. All models (MMPL, ABMIL, and ZoomMIL) were evaluated under identical conditions using a single CPU thread. The theoretical complexity analysis uses standard variables, with $N$ denoting the number of patches and $d$ the feature dimension.
>
> In terms of throughput, MMPL achieves the highest performance because classification is applied only to the top-k retrieved patches. For FLOPs measured at $N=10,000$ patches, MMPL is the most efficient at 10.71 GFLOPs, compared with 15.75 GFLOPs for ABMIL and 31.73 GFLOPs for ZoomMIL. The observed FLOPs ratio (MMPL:ABMIL:ZoomMIL ≈ 1:1.5:3) aligns closely with the theoretical complexity ratio of 1:1.5:2.5, which supports the architectural efficiency of our method.

---

> ### Author Response · Authors · 2025-11-27
>
> > The method depends on prototype count M, the queue size K, and the top-k allocation vector Ij. Only limited ablation is shown.
> How sensitive is performance to M?
> >
>
> The ablation studies for the CAMELYON16 dataset on the number of top-k patches and the number of prototypes are presented in Supplementary Table 3 and Figure 1. Building on these initial analyses, we extended the ablation study to the remaining datasets by examining the effect of the queue size, the number of top-k patches, and the number of prototypes using ResNet50.
>
> ### Feature Queue Size
> | Dataset | Camelyon16 |  |  |  |  | PANDA (K) | PANDA (R) |  |  | TCGA (5-fold-cv) |  |  |
> | --- | --- | --- | --- | --- | --- | --- | --- | --- | --- | --- | --- | --- |
> | queue_size | Acc | WF1 | AUC |  | queue_size | Kappa | Kappa |  | queue_size | Acc | WF1 | AUC |
> | 85000 | 0.854 | 0.853 | 0.880 |  | 25000 | 0.831 | 0.781 |  | **150000** | **0.882** | **0.885** | 0.924 |
> | 90000 | 0.853 | 0.851 | 0.876 |  | 50000 | 0.805 | 0.785 |  | 200000 | 0.875 | 0.875 | **0.925** |
> | **100000** | **0.885** | **0.885** | **0.924** |  | 70000 | 0.800 | 0.799 |  | 250000 | 0.879 | 0.878 | **0.925** |
> | 150000 | 0.795 | 0.793 | 0.798 |  | 90000 | 0.790 | 0.789 |  |  |  |  |  |
> | 200000 | 0.785 | 0.781 | 0.768 |  | **100000** | **0.838** | **0.841** |  |  |  |  |  |
> |  |  |  |  |  | 150000 | 0.826 | 0.802 |  |  |  |  |  |
>
> ### Top-k
>
> | Dataset | PANDA (K) | PANDA (R) |  | TCGA |  |  |
> | --- | --- | --- | --- | --- | --- | --- |
> | top_k | Kappa | Kappa | top_k | Acc | WF1 | AUC |
> | 80 | 0.817 | 0.759 | 80 | 0.871 | 0.870 | 0.929 |
> | **100** | **0.836** | **0.820** | **100** | **0.876** | **0.875** | **0.930** |
> | 120 | 0.820 | 0.812 | 120 | 0.856 | 0.857 | 0.923 |
> | 150 | 0.830 | 0.768 | 150 | 0.871 | 0.871 | 0.908 |
>
> ### Number of Prototypes
> | Dataset | PANDA (K) | PANDA (R) |  | TCGA |  |  |
> | --- | --- | --- | --- | --- | --- | --- |
> | num_proto | Kappa | Kappa | num_proto | Acc | WF1 | AUC |
> | 3 | 0.814 | 0.775 | 3 | 0.780 | 0.780 | 0.890 |
> | 4 | 0.847 | 0.796 | 4 | 0.837 | 0.837 | 0.898 |
> | **5** | **0.875** | **0.836** | 5 | 0.847 | 0.847 | 0.914 |
> | 10 | 0.802 | 0.740 | 10 | 0.828 | 0.828 | 0.889 |
> | 15 | 0.836 | 0.820 | **15** | **0.876** | **0.875** | **0.930** |
> | 17 | 0.818 | 0.802 | 17 | 0.866 | 0.866 | 0.923 |
>
> The results suggest that the appropriate number of prototypes $M$ may depend on the domain and task. On PANDA, we observe that choices of $M$ other than 15 can perform better than $M=15$, indicating that there is room for further gains by more carefully tuning $M$ for this dataset.
>
> > Theoretical justification or failure modes. While OT with uniform marginals mitigates collapse, the paper lacks a short formal discussion of when prototype assignments might still be suboptimal (e.g., extremely class-imbalanced slides) and how the method handles rare morphologies.
> Despite OT uniform marginals, is there a risk that some prototypes remain unused? Provide the distribution of prototype assignments (counts per prototype) and show whether OT enforcement truly yields balanced semantic coverage.
> >
>
> ### The distribution of prototype assignments
>
> | Prototypes | P0 | P1 | P2 | P3 | P4 | P5 | P6 | P7 | P8 | P9 | P10 | P11 | P12 | P13 | P14 |
> | --- | --- | --- | --- | --- | --- | --- | --- | --- | --- | --- | --- | --- | --- | --- | --- |
> | **Count** | 17,619 | 57,956 | 700 | 37,089 | 4,990 | 449 | 10,450 | 116,425 | 158,530 | 2,269 | 66,009 | 23,908 | 353,527 | 184,660 | 309,143 |
>
> We agree that clustering-based methods are vulnerable to dead centroids. Although OT with uniform marginals is less prone to collapse than k-means, extremely imbalanced WSI patch distributions can still lead to some prototypes receiving very few or even zero assignments on a given slide.
>
> To address this risk of unused prototypes, our methodology explicitly disentangles prototype learning from slide-level classification. OT is used primarily to update prototype locations over the entire feature queue, rather than to define the final MIL decision rule. Even if, within a particular imbalanced WSI, some prototypes temporarily receive no OT assignments, they are likely to be assigned patches from other WSIs in subsequent batches, which allows them to continue adapting instead of becoming permanently inactive.
>
> For MIL prediction, we retrieve the top-k most similar patches for every prototype, independent of the instantaneous OT assignment counts. This design prevents transient OT instabilities from propagating directly into the classification stage and ensures that even rarely assigned prototypes remain involved in the decision process.
>
> Empirically, we also measure the nearest-neighbor assignment distribution across the entire test set. We observe that every prototype is selected by at least one feature, confirming that our disentangled design effectively mitigates prototype collapse and maintains broad semantic coverage.

---

> ### Author Response · Authors · 2025-11-27
>
> > Provide clearer details and results comparing the two-step frozen-backbone training to full end-to-end training. If possible, Could you provide a short expert evaluation or annotation to show prototypes align with meaningful histology concepts?
> >
>
> Thank you for requesting a comparison between the end-to-end and two-step approaches. Because images cannot be attached directly in the system, we included the visualization in the revised manuscript. Specifically, we added a localization map comparison for ResNet50-trained models on the CAMELYON16 dataset (Supplementary Figure 6). The visualization highlights, for each patch, the prototype with the highest similarity score. From left to right, it presents the original slide, the two-step approach, and the end-to-end (E2E) approach.
>
> In the original slide, the annotated region is outlined in yellow. In both the two-step and E2E localization maps, the color spectrum represents prototype importance, with colors progressing from purple to yellow indicating higher importance.
>
> The E2E approach shows a markedly closer correspondence to the annotated regions than the two-step method. Notably, it captures the smaller annotated area on the left side of the slide, which the two-step approach fails to highlight. This improvement is likely attributable to the vision encoder being trained exclusively on top-k patches in the E2E pipeline, which concentrates learning on regions with a higher proportion of annotated patches.
>
> Furthermore, we computed the overlap ratio between prototype-assigned patches and tumor annotations. The E2E method achieves a higher maximum prototype overlap ratio on the CAMELYON16 dataset, reaching 89.19% compared to 84.03% for the two-step method. This quantitative gain aligns with the qualitative differences observed in the localization maps and explains the superior performance of the E2E method reported in Supplementary Table 7 relative to the two-step method reported in Supplementary Table 1.
>
> > The MMPL architecture diagram is not aesthetically pleasing, except for the defect of misaligned box lines at first glance. Some symbols do not appear in the main text and are not explained in the captions.
> >
>
> We truly appreciate your careful review of our figures and symbols, which has helped us enhance the clarity of our visualizations. We have addressed the points regarding the figure and symbols in the revision.
>
> ---
>
> We appreciate your thoughtful review and have incorporated your suggestions throughout the revised manuscript. We believe the additional analyses respond directly to your comments, and we welcome any further guidance you may have.

---

### Official Review · Reviewer_68kJ · 2025-11-08

**Soundness:** 3
**Presentation:** 3
**Contribution:** 2
**Rating:** 4
**Confidence:** 5

**Summary:**

This work introduces MMPL, a prototype-based method that searches whole-slide images for only the most informative patches to make a slide-level decision. MMPL works by learning a small set of global “morphological” prototypes with a momentum encoder and queue, assigning patch embeddings to prototypes via Sinkhorn-Knopp optimal transport, and then retrieving the top-k relevant patches per prototype to compute the prediction using just those patches. The proposed novelty is the combination of prototype learning with a memory queue for stable training and a dynamic allocation of retrieved patches per prototype. Experiments span CAMELYON16, PANDA (K/R), and TCGA tasks, comparing against MIL baselines such as ABMIL, DSMIL, CLAM, TransMIL, etc., and also evaluate different encoders including ResNet-50 and pathology foundation models (UNI, CONCH, and GigaPath).

**Strengths:**

- Lots of ablations: top-k selection, the effect of the queue and prototype components, uniform versus dynamic per-prototype selection, prototype assignment via Sinkhorn versus K-means, and backbone choices.
- Presentation of the paper is good and math is sound. Method extends SK-OT for pathology and is overall sound (working both unsupervised and supervised).

**Weaknesses:**

- Very limited experiments and tasks. Only three datasets are evaluated (C16, PANDA, TCGA-Lung) and are only classification (no survival tasks). While experimental design on ablating performance of MMPL is sound, most of the tasks that MPPL are evaluated on are a bit simple. It would be interesting to evaluate on a greater range of tasks.
- Unclear if number of prototypes in MMPL are fixed versus adaptive, e.g. if the number of clusters can change. Why is the number of prototypes fixed to 15? Interpretability of MMPL clusters could also show deeper insights (comparison with PANTHER in learning quality of prototypes).
- Hard to understand where performance of supervised versus unsupervised MMPL is evaluated.
- Technically, this work addresses the problem of learning better prototypes for weakly-supervised classification tasks in pathology. The work generally extends the idea of SK-OT to the pathology domain, and though mostly applied, has good empirical experimentation of the technical components. For better impact, I would like to see more interpretability and evaluation on more diverse tasks as there are many MIL architectures that succeed on the evaluated tasks.

**Questions:**

N/A

---

> ### Author Response · Authors · 2025-11-27
>
> We sincerely thank the reviewer for the highly insightful feedback, which recognized the soundness of our methodology and the rigor of our ablation studies.
>
> ---
>
> > Very limited experiments and tasks. Only evaluated on three datasets (CAMELYON16, PANDA, TCGA-Lung) and only on classification, without more complex tasks such as survival analysis.
> >
>
> We appreciate your valuable feedback regarding the importance of task diversity, such as incorporating prognosis or survival analysis. We agree that exploring these complex endpoints is crucial. In fact, our study specifically focused on classification tasks because our primary objective is to learn explicit and localized pathological signals in sparse tumor regions.
>
> Classification tasks provide a clear, explicit signal crucial for validating our prototype-based explicit feature selection mechanism, especially in cases where the disease is rare within the image. Survival analysis often focuses on complex, patient-level noise and multi-modal features, which diverges from our core goal of patch-level pathological localization. While MMPL can be easily adapted to these tasks, we strategically prioritized the explicit signal task.
>
> > Unclear if number of prototypes in MMPL are fixed versus adaptive, e.g. if the number of clusters can change. Why is the number of prototypes fixed to 15?
> >
>
> Thank you for raising the concern about the need for deeper insights into the prototypes. We confirm that our framework is architecturally flexible to accommodate both static (fixed) and dynamic (adaptive during training) prototype selection mechanisms. However, for the scope of this work, we opted for a fixed number of prototypes for key reasons:
>
> - A fixed K ensures a stable and controlled setting necessary for our rigorous ablation studies.
> - We empirically validated the influence of the prototype count, $K$, using a fixed-count ablation across a range of values, finding that $K=15$ provides the optimal performance and empirically justifies our static choice.
>
> As shown in Figure 1(c) of the Supplementary Material (prototype ablation study), $K=15$ was deemed appropriate for this work. This selection allows us to use a suitable, representative number of prototypes for the entire dataset, balancing the potential for better, more granular performance at other values with the practical need for a fixed configuration across our main experiments.
>
> This fixed approach aligns with common practice in prototype-based MIL papers [1–3], where the number of prototypes or prototypical parts is treated as a task-dependent hyperparameter and chosen empirically for a given dataset.

---

> ### Author Response · Authors · 2025-11-27
>
> > Interpretability of MMPL clusters could also show deeper insights (comparison with PANTHER in learning quality of prototypes).
> >
>
> ### Panther (annotated ratio per-prototype)
>
> | P15 | P11 | P9 | P0 | P8 | P6 | P14 | P7 | P2 | P13 | P12 | P1 | P5 | P4 | P3 | P10 |
> | --- | --- | --- | --- | --- | --- | --- | --- | --- | --- | --- | --- | --- | --- | --- | --- |
> | 19.85% | 11.81% | 8.65% | 4.50% | 1.94% | 1.87% | 1.58% | 0.62% | 0.35% | 0.29% | 0.18% | 0.13% | 0.11% | 0.03% | 0.02% | 0.00% |
>
> ### MMPL (annotated ratio per-prototype)
>
> | P11 | P6 | P10 | P2 | P14 | P1 | P0 | P8 | P4 | P9 | P3 | P13 | P5 | P12 | P7 |
> | --- | --- | --- | --- | --- | --- | --- | --- | --- | --- | --- | --- | --- | --- | --- |
> | 84.03% | 31.94% | 11.06% | 2.57% | 1.80% | 1.56% | 1.45% | 1.39% | 1.36% | 1.10% | 0.98% | 0.62% | 0.45% | 0.37% | 0.30% |
>
> To evaluate the learning quality of each prototype, we measured the proportion of retrieved patches that correspond to the annotated tumor region. Specifically, for each prototype, we calculated the ratio of patches within the top-k retrieved set that were annotated as tumor.
>
> Based on the provided tables showing the annotation overlap ratios for the Camelyon16 test dataset, our learnable prototype method significantly shows stronger ability to cluster tumor-relevant patches than the PANTHER prototype method.
>
> ### Comparison of Maximum Overlap:
>
> - The highest annotation overlap ratio achieved by a single PANTHER prototype is 19.85% (Prototype 15).
> - In sharp contrast, the highest overlap ratio achieved by a prototype in MMPL(ours) is 84.03% (Prototype 11).
> - This dramatic difference clearly demonstrates that our model's prototypes are much more effective at clustering patches that are highly relevant to tumor detection (i.e., patches within the annotated tumor region).
>
> ### Comparison of Average Overlap:
>
> - The average overlap ratio across all PANTHER prototypes is 3.25%.
> - The average overlap ratio across all prototypes in MMPL(ours) is 9.40%.
> - This indicates that, on average, our model's prototypes are substantially better at capturing the annotated tumor area across the entire set of clusters compared to the PANTHER method.
>
> > Hard to understand where performance of supervised versus unsupervised MMPL is evaluated.
> >
>
> We acknowledge the need for precise methodological terminology, and accordingly, the term previously referred to as 'unsupervised' was revised to 'self-supervised' to reflect its more appropriate classification in the revision.
>
> Our method operates under a **Joint Objective of Supervised and Self-Supervised Learning paradigm**. This is achieved by combining two distinct objectives in a joint loss function,
>
> $$
> L=L_{CE}+\lambda L_{prototype},
> $$
>
> combines a supervised classification loss ($L_{CE}$) and a self-supervised prototype learning loss ($L_{prototype}$).
>
> - The supervised component is responsible for achieving the final slide-level diagnosis or classification. It utilizes the Gated Attention pooling layer and linear classifier to aggregate the final Top-k retrieved patch features and produce the slide-level prediction. This prediction is trained against the ground-truth slide label using the $L_{CE}$ loss.
> - The self-supervised component focuses on learning prototypes without reliance on the slide-level label. This process involves calculating the pseudo-labels matrix by performing the Optimal Transport (OT) assignment of features from the Feature Queue to the learnable prototypes. This assignment serves as the supervision signal for the prototype learning loss ($L_{prototype}$), which prevents prototype collapse and ensures diverse feature representation.
>
> ---
>
> Thank you for the constructive feedback that informed these revisions. We believe that the updated analyses resolve the concerns you raised, and we would be glad to address any remaining questions or recommendations.
>
> [1] Yu J-G et al. Prototypical multiple instance learning for predicting lymph node metastasis of breast cancer from whole-slide pathological images. \
> [2] Sun S et al. Prototype-Based Multiple Instance Learning for Gigapixel Whole Slide Image Classification. \
> [3] Xia et al. Learnable Prototype-Guided Multiple Instance Learning for Detecting Tertiary Lymphoid Structures in Multi-Cancer Whole-Slide Pathological Images

---

### Author Response · Authors · 2025-12-03
**Summary for the Area Chair and Reviewers**

We are deeply grateful to the Area Chair for the considerable time and effort devoted to overseeing and assessing our work. We also sincerely thank the reviewers for their thorough evaluation and highly constructive feedback on our submission. The reviews consistently recognized the novelty, technical soundness, and practical significance of our Momentum Morphological Prototype Learning (MMPL) framework.
Based on the insightful comments, we conducted additional experiments, performed further analyses, and improved the clarity of the manuscript. We believe that all concerns have been resolved.

The reviewers' comments collectively highlight the novelty, technical soundness, and strong empirical support of MMPL.

- **Novel Paradigm and Efficiency:** MMPL presents a fresh and impactful approach by reformulating WSI diagnosis as an efficient prototype-driven multi-vector retrieval problem. (@6sEU, @kdfC, @tnwR)
- **Technical Soundness and Stability:** The integration of Optimal Transport (OT) for diverse, non-collapsing prototypes and a momentum-based feature queue (with EMA updates) for stable learning is regarded as technically sound and well-motivated. (@68kJ, @6sEU, @tnwR)
- **In-Depth Ablation Studies and Experimental Validation:** Multiple reviewers emphasized that the extensive ablation studies (e.g., top-k, queue, prototypes, OT vs. K-means, dynamic vs. uniform retrieval) and experiments across three public WSI benchmarks (CAMELYON16, PANDA, TCGA) provide substantial empirical evidence for the effectiveness of MMPL. (@68kJ, @6sEU, @KdfC)

We carefully addressed the major concerns and remaining questions raised by the reviewers.

1. Computational Costs (@6sEU, @kdfC, @tnwR)

    We clarified that the main efficiency benefit of our end-to-end (E2E) pipeline comes from the Top-K retrieval mechanism, which reduces the number of patches processed by the aggregation network and enables memory-efficient training of the vision encoder on only the top-k retrieved patches. In our rebuttal, we additionally reported  controlled measurements of throughput, memory usage, and FLOPs, showing that **MMPL achieves higher slide-level throughput and lower FLOPs than strong baselines such as ABMIL and ZoomMIL on the same hardware**.

2. Prototype Interpretability and Prototype Usage (@68kJ, @6sEU)

    We provided additional analyses to address both the interpretability of prototypes and concerns about dead centroids and rare morphologies in the OT-based assignment:

    - **Interpretability:** Comparing MMPL with PANTHER, we quantitatively demonstrated that MMPL’s prototypes capture meaningful pathological signals. In particular, one prototype attains **an 84.03% overlap with tumor-annotated regions on CAMELYON16**, and the average overlap across prototypes is also substantially higher than that of PANTHER.
    - **Stability and Usage:** We reported the distribution of prototype assignments and showed that, by decoupling OT-based prototype learning from the MIL prediction stage and operating over a large feature queue, **no prototype degenerates into a dead centroid in practice**. In addition, at prediction time we retrieve the top-k most similar patches for every prototype, so even rarely assigned prototypes remain involved in the slide-level decision.
3. More Ablation Studies and Hyperparameter Sensitivity (@6sEU, @tnwR)

    We expanded the scope of our ablation studies in response to the reviewers’ requests for a more systematic analysis of key hyperparameters:

    - **Feature Queue Size:** We added ablations on the feature queue size using the ResNet50 backbone for the PANDA and TCGA datasets, complementing the original analysis on CAMELYON16.
    - **Top-K and Number of Prototypes:** We extended the ablation studies on the Top-K value and the number of prototypes to PANDA and TCGA, in addition to the initial results on CAMELYON16. These results illustrate how the optimal prototype count and retrieval depth can vary by dataset and task.

---

Beyond these shared themes, we have also responded point-by-point to all remaining reviewer-specific questions and incorporated the corresponding revisions into the manuscript.

---

### Meta-Review · Area_Chair_hRwm · 2025-12-13

**Summary:**

The authors propose MMPL for Whole Slide Image (WSI) classification.  The method redefines WSI diagnosis as a retrieval task, using a learnable set of prototypes to identify and select the top-$k$ most relevant patches for downstream classification. To achieve this, the optimization target combines the supervised loss with a self-supervised approach to learn a global prototype. Key technical ideas include the use of Optimal Transport (OT) with Sinkhorn-Knopp to assign patches to prototypes (preventing collapse) and a momentum encoder with a feature queue to stabilize training. The authors claim this approach improves performance and computational efficiency by processing only a subset of informative patches during the aggregation stage.

**Reviewer Concerns:**

1. The authors clarified that the efficiency gains of MMPL are primarily in memory efficiency during end-to-end training (by backpropagating only through top-$k$ patches) and computational load during the aggregation stage. However, despite the clarifications, reviewers remained concerned that the method does not address the primary bottleneck of WSI analysis: the time-consuming extraction for all patches.

2. The authors provided extensive new ablations on feature queue size, top-$k$, and prototype counts across PANDA and TCGA datasets.

there are more outstanding concerns:

1. Limited novelty. The idea of morphological prototyping has been well-acknowledged in this field, for example, in PANTHER. The incremental advancement with OT loss is not that attractive.

2. Limited task scope: The evaluation remains focused solely on classification. The authors declined the request for a survival analysis (prognosis) to demonstrate robustness on more complex, noisy tasks, leaving the method's efficacy on non-classification WSI tasks unproven.

3. Baselines. Even with updated experiments, the IBMIL baseline exhibited unusually high variance compared to other methods. Also, the baselines do not reflect the state-of-the-art techniques in 2024/2025.

4. generalization. Although cam16 and panda are widely used benchmarks, more out-of-domain datasets should be validated (train on one cohort and validate on an independent cohort).

**Reviewer Scores:**

All reviewers reached a consensus that this paper is 4. I believe they will maintain their scores after considering the rebuttal.

---

### Decision · Program_Chairs · 2026-01-26

Reject